# Hypersensitive detection of single millimeter vascular emboli from adhesive in vivo

Ruihan Liu ®[1,5], Shuo Li[1,5], Xingyu Gao[1,5], Quan Zou[1], Gang Shu[2], Cai Zhang[2], Jinbin Pan ®[3], Xiaoyuan Chen ®[4] ✉ & Shao-Kai Sun ®[1] ✉

Surgical adhesives are widely used in clinical practice but pose a significant risk of severe vascular embolism complications. Nevertheless, there are currently no non-invasive direct methods for precise detection of detached emboli. Herein, we show a CT-visualized method for hypersensitive detection of single millimeter vascular emboli from adhesive in vivo by simply doping BiOCl into surgical adhesives. As proof of concept, BiOCl-BioGlue with excellent CT imaging capability is fabricated and applied to repair ruptured vessels and liver in male rats. The location, morphology, and degradation process of BiOCl-BioGlue can be dynamically monitored by CT imaging for 42 days, and pulmonary emboli caused by BiOCl-BioGlue, with sizes as small as 1.2 mm, are successfully detected. Additionally, the high K-edge of Bi enables precise detection of pulmonary emboli in spectral CT imaging, unaffected by confounding calcifications. The proposed non-invasive detection strategy for adhesive emboli significantly enhances the biosafety of surgical adhesives.

Surgical adhesives, such as the FDA-approved BioGlue, TISSEEL, Progel, and PreveLeak, offer several advantages, including reducing operative time, decreasing bleeding, minimizing tissue damage, lowering infection risk, enhancing patient comfort, and diminishing scar formation. They are widely utilized in clinical practice for hemostasis, reinforcement of suture lines, and the repair of fragile tissues, significantly enhancing the safety and efficacy of surgical procedures[1–3]. Despite significant clinical benefits, the leakage and detachment of surgical adhesives at suture lines may lead to severe complication of vascular embolism[4,5]. As the most extensively used surgical adhesive, BioGlue poses a risk of generating adhesive-derived emboli during and after surgery, and embolism commonly occurs in pulmonary artery, coronary artery, and cerebral vessels in clinic[6–10]. Adhesive-derived emboli with different sizes (ranging from 1.0 mm to several millimeters) may affect tissue perfusion, leading to ischemia and necrosis, and can even result in cerebral infarction and myocardial infarction, potentially endangering life. Besides, adhesive-derived emboli cannot

be effectively treated by conventional anticoagulant or thrombolytic therapies and typically require interventional or surgical removal. Thus, early and accurate detection of the embolus is crucial for timely treatment of vascular embolism complication.

Computed tomography angiography (CTA) and Digital subtraction angiography (DSA) are most used imaging methods for detecting vascular embolism in clinic, which diagnoses embolism by detecting obstructed or interrupted contrast agent flow[11–13]. As contrast agent injection is required, they are typically performed only when patients exhibit clinical symptoms, resulting in difficulties in early detection and missing the optimal treatment opportunities. Moreover, CTA and DSA are indirect methods for embolism detection, which is unable to directly visualize embolus or classify their origins. In recent years, intravascular ultrasound (IVUS) and optical coherence tomography (OCT) are employed for directly visualizing BioGlue-induced embolus successfully[14,15]. These two methods are invasive, have limited penetration depth, and are not commonly utilized in clinical examinations.

[1]School of Medical Imaging, Division of Medical Technology, Tianjin Key Laboratory of Functional Imaging, Tianjin Medical University, Tianjin, China. [2]Department of Radiology, Tianjin Medical University Cancer Institute and Hospital, National Clinical Research Centre of Cancer, Tianjin Clinical Research Center for Cancer, Tianjin, China. [3]Department of Radiology, Tianjin Key Laboratory of Functional Imaging, Tianjin Medical University General Hospital, Tianjin, China. [4]Shandong Provincial Key Laboratory of Precision Oncology, Shandong Cancer Hospital and Institute, Shandong First Medical University and Shandong Academy of Medical Sciences, Jinan, China. [5]These authors contributed equally: Ruihan Liu, Shuo Li, Xingyu Gao. ✉ e-mail: chen9647@gmail.com; shaokaisun@tmu.edu.cn

Therefore, it is of great significance to develop new methods for the noninvasively precise detection of adhesive-induced vascular embolism.

Incorporating imaging units into adhesives or gels is a commonly employed strategy for visualizing them through various imaging techniques like computed tomography (CT) imaging[16–19], magnetic resonance imaging[20], nuclear imaging[17,21], ultrasound imaging[22–24], and photoacoustic imaging[25–27]. These approaches have been extensively utilized to detect the location, morphology, and degradation behavior of adhesives and gels, and has found widespread application in tissue engineering, disease diagnosis, drug delivery, and immunotherapy. Inspired by these advanced developments, incorporating imaging probes into adhesives provides a valuable opportunity for the non-invasively precise detection of adhesive-induced emboli. However, there are currently no relevant studies to our best knowledge.

In this work, we presented an ultrasimple strategy to endow surgical adhesives with CT visualization capability for hypersensitive detection of single millimeter vascular emboli from adhesive in vivo for the first time (Fig. 1). The CT-visualized adhesive was synthesized by incorporating amorphous BiOCl with excellent X-ray absorption property and good compatibility with adhesive via a straightforward and universal method. The introduction of BiOCl allowed various adhesives to achieve remarkable CT imaging capability without compromising their gelation rate and adhesion strength. Using BiOCl-doped adhesive as an example, we conducted a systematic study on wound adhesion, adhesive monitoring, and detached embolus detection in vivo. The Bi-BioGlue demonstrated high efficacy in repairing ruptured blood vessel and liver, with its location and degradation dynamically monitored by CT imaging over a period of 42 days. Furthermore, the embolus as small as 1.2 mm, derived from Bi-BioGlue, was readily detected by CT imaging in a pulmonary embolism model. Additionally, due to the high K-edge energy value of Bi, the embolus was precisely distinguished from calcifications within the lungs using spectral CT imaging. Our study provided a noninvasive, precise and universal method for the early detection of adhesive-associated embolus, thereby significantly improving the biosafety of surgical adhesives.

## Results

### Synthesis and Characterization of CT-visualized Adhesives

Among various imaging modalities, CT imaging is routinely used in clinical settings due to its deep tissue penetration, high density resolution, and fast imaging speed, making it exceptionally suitable for the detection of emboli[28,29]. However, due to the small size of emboli and their uncertain distribution within the body, the CT imaging contrast agent used to label surgical adhesives must possess high sensitivity and good biocompatibility. Compared to the widely used iodine-based CT contrast agents with lower sensitivity, elements with high atomic numbers and greater X-ray absorption capabilities are increasingly utilized to develop CT contrast agents with enhanced sensitivity. Notably, as bismuth (Bi) element possess highest atomic number in all non-radioactive elements, thus exhibiting superior X ray attenuation ability[30]. In addition, Bi, the most cost-effective high atomic number element for constructing CT imaging probes[31–38], offers good biocompatibility and many Bi-drugs, like colloidal bismuth subcitrate, ranitidine bismuth citrate, and bismuth subsalicylate are widely used in basic research and in clinic[39–45]. Therefore, it is an ideal choice to introduce Bi-based probe into surgical adhesives, combined with CT imaging for early and precise detection of embolism complication.

BioGlue is composed of purified bovine serum albumin (BSA, 45%, w/v) and glutaraldehyde (GA, 10%, w/v), which are mixed in a dual-barrel syringe and applied to the surgical site. The principle behind BioGlue is that glutaraldehyde forms a stable seal by covalently linking to the lysine amino acid fragment in BSA, and it can also connect to lysine residues in other proteins in the tissue, resulting in excellent

tissue adhesion (Fig. 2a). The strategy of incorporating imaging probes into surgical adhesives requires that the probes are evenly distributed within the adhesive and have an appropriate labeling duration to ensure optimal CT imaging results and an extended imaging time window. We selected several Bi-based imaging probes with varying physicochemical properties, including completely water-soluble Bi-DTPA small molecules (Fig. S1a), BiOCl flocculent precipitates (Fig. S1b), fully hydrophobic $Bi_2S_3$ and $Bi_2O_3$ powders, to investigate their effects on labeling the most commonly used clinical adhesive, BioGlue. Bi-DTPA was synthesized according to our previously reported method[46] (Fig. S1c), BiOCl, composed of aggregated nanoparticles with poor water solubility, was fabricated using a simple hydrolysis method (Fig. S1d, e), and $Bi_2O_3$ and $Bi_2S_3$ were obtained through commercial sources. The Bi-based BioGlue was synthesized facilely by mixing Bi-based probes with a BSA solution, followed by the addition of GA. Bi-based BioGlue and BioGlue share the same crosslinking mechanism, which is based on the covalent interaction between BSA and GA. The only difference lies in the composition of the protein component: Bi-based BioGlue incorporates BSA doped with Bi-based imaging probes, whereas BioGlue utilizes native BSA (Fig. 2a). The GA can cross-link with BSA via Schiff base formation between its aldehyde groups and amino groups in BSA. All four types of Bi-based probes were doped into BSA solution and successfully formed gels upon incorporation with GA (Fig. 2b). CT images revealed that all four Bi-based BioGlue samples exhibited significant contrast enhancement compared to the blank BioGlue. However, the contrast enhancement in the BioGlue doped with $Bi_2S_3$ and $Bi_2O_3$ was notably uneven, attributed to the strong hydrophobicity of $Bi_2S_3$ and $Bi_2O_3$ materials, causing their uneven distribution within the adhesive (Fig. 2b, c). In contrast, the uniform CT enhancement indicated that BiOCl and Bi-DTPA can be evenly dispersed in BioGlue. However, the CT value of Bi-DTPA-BioGlue decreased sharply within 5 h when soaked in water, due to the small molecular size and good water solubility of Bi-DTPA, which allowed it to rapidly diffuse from BioGlue into the surrounding solution. In the same conditions, the CT value of BiOCl-BioGlue remained stable over 24 h. These results suggested that, considering both labeling uniformity and persistence, BiOCl provided the optimal labeling effect for the adhesive (Fig. 2d).

Bi doping concentration was systematically optimized to assess its effect on the CT imaging performance of Bi-BioGlue. Increasing the Bi content enhanced the sensitivity of CT detection; however, excessive doping led to undesirable changes in the colloidal properties and raised potential concerns about biosafety. The result showed that the CT value of bulk Bi-BioGlue gradually increased with rising Bi concentrations. When the Bi concentration reached 105 mM, the CT value of Bi-BioGlue reached approximately 659 HU, yielding a favorable signal-to-background ratio compared to normal tissues, which typically exhibit CT values in the range of 40-80 HU. It should be noted that maintaining a high CT value in bulk Bi-BioGlue relative to soft tissue is essential, as the formation of very small BioGlue emboli will lead to a certain degree of CT signal attenuation due to partial volume effects. Therefore, 105 mM was selected as the optimal Bi doping concentration for the following studies.

Besides BioGlue, BiOCl was further attempted to be incorporated into other types of adhesives, including gelatin and sodium alginate adhesives[47–49]. All BiOCl-doped gels exhibited the same gelation process as the corresponding blank gels and demonstrated similar CT contrast enhancement effects as Bi-BioGlue, confirming the good versatility of the BiOCl doping strategy in surgical adhesives (Fig. 2e and Fig. S2).

The amide I band vibrations at 1655 cm$^{-1}$ and amide II band vibrations at 1540 cm$^{-1}$ in BSA were both observed in the FTIR spectrum of the BioGlue, Bi-BioGlue, and characteristic O = CH- stretching vibration at 1710 cm$^{-1}$ for GA disappeared after the crosslinking reaction with BSA, indicating the successful synthesis of BioGlue and Bi-

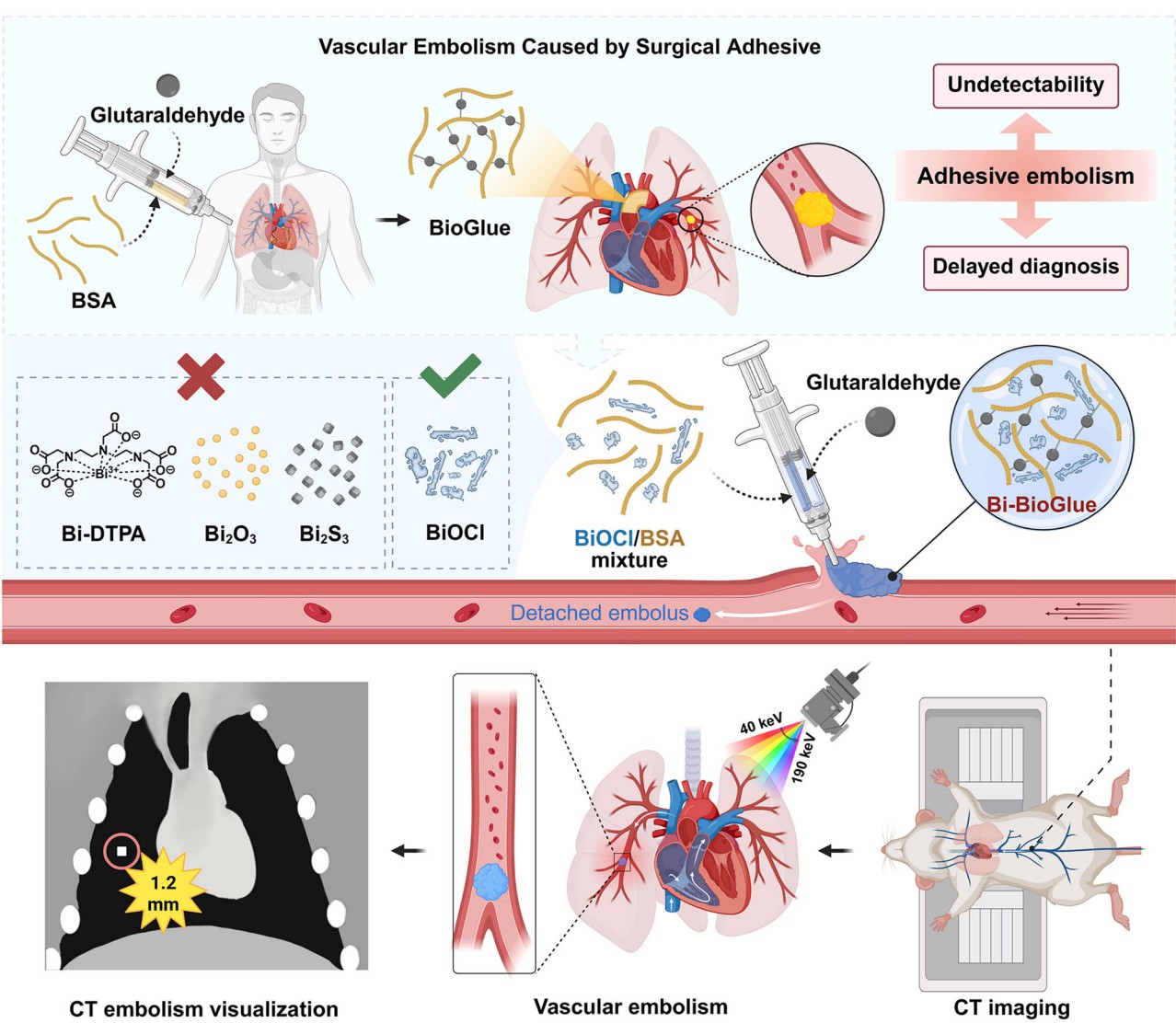

**Fig. 1 | Schematic illustration of CT-visualized Bi-BioGlue capable of hypersensitive detection of single millimeter vascular emboli from adhesive in vivo.** The elements were created in BioRender. Sun, S. (2025) https://BioRender.com/aqjvl21.

BioGlue (Fig. 2f)[50]. Elemental mapping images demonstrated the presence and uniform distribution of Cl (orange), Bi (yellow), and S (purple) in the freeze-dried Bi-BioGlue (Fig. 2g), proving the homogeneous distribution of BiOCl in as-prepared Bi-BioGlue. Furthermore, the synthesis strategy can be extended to the preparation of other metal (Hf/Ta/Yb)-doped BioGlue (Fig. S3). Dynamic oscillation time sweep measurements showed that exhibit similar G′/G″ ratios, indicating that doping with metal elements does not affect the rheological properties of the adhesives (Fig. S4). The crosslinking kinetics of both BioGlue and Bi-BioGlue by rheological analysis demonstrated that upon mixing, the BSA and glutaraldehyde components exhibited similarly rapid crosslinking kinetics, forming an opaque white solid within seconds (Fig. 2h). The Bi-BSA mixture and GA could be easily extruded from the needle to form Bi-BioGlue with customizable pattern (e.g., TMU) with good CT imaging capacity (Fig. 2i). In addition, the morphological changes of Bi-BioGlue after incubation in various simulated physiological media over different time periods and the release of Bi ions were systematically studied. The experimental results demonstrated that Bi-BioGlue maintained a stable morphology throughout the monitoring period, with negligible $Bi^{3+}$ release (Fig. S5, S6). Further XRD analysis showed that BiOCl, whether incorporated into Bi-BioGlue or not, retained its crystalline structure across all media

without noticeable changes (Fig. S7). These findings indicate that BiOCl exhibits efficient labeling within the BioGlue matrix, and the resulting Bi-BioGlue possesses excellent physicochemical stability.

## In Vitro Conventional/Spectral CT Imaging
For comparison, clinically used iodine-based contrast agents was also doped into BioGlue (I-BioGlue). Though CT intensity of both Bi-BioGlue and I-BioGlue was linear enhancement with the increasement of concentration, the Bi-BioGlue yielded a more pronounced CT image brightness and higher CT intensity compared to I-BioGlue on same Bi/I-doped concentration at commonly used tube voltage in clinic (120 kV) (Fig. 2e and Fig. S8). In addition, since clinical iodine-based contrast agents are small molecules with excellent water solubility, I-BioGlue exhibited a burst release behavior of imaging probes similar to that of Bi-DTPA-BioGlue when immersed in water. These results indicated that clinical iodine-based contrast agents are unsuitable for labeling surgical adhesives due to their relatively poor sensitivity and labeling persistence (Fig. S9).

Traditional CT imaging uses a single X-ray tube emitting mixed-energy X-rays, producing images based on the absorption of these mixed-energy X-rays by the scanned object. In contrast, spectral CT is an advanced imaging technology that utilizes two X-ray tubes

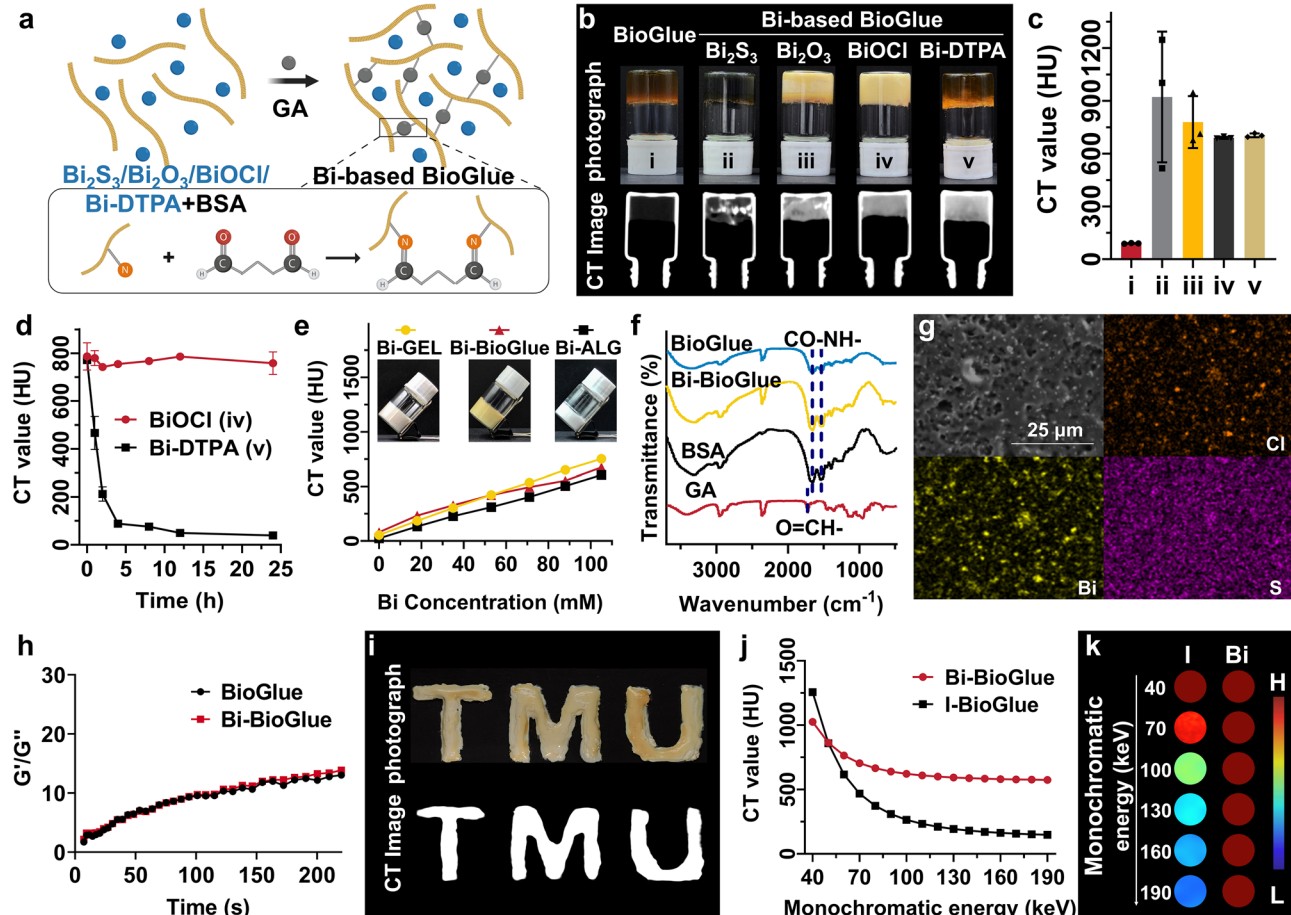

**Fig. 2 | Synthesis and characterization of CT-visualized adhesives. a** Synthetic scheme of the Bi-doping BioGlue through the GA-induced cross-linking method. **b** Photographs and corresponding CT images of conventional BioGlue (i) and $Bi_2S_3$ (ii), $Bi_2O_3$ (iii), BiOCl (iv) or Bi-DTPA (v) -BioGlue with the Bi concentration of 105 mM. **c** Corresponding CT values of conventional BioGlue and Bi-based BioGlue ($n = 3$ technical replicates; mean ± SD). **d** CT value change curves for BiOCl-BioGlue and Bi-DTPA-BioGlue in water for 24 h ($n = 3$ technical replicates; mean ± SD). **e** CT value change curves of Chitosan-gelatin, BioGlue and sodium alginate-$Ca^{2+}$ adhesives after doping of different concentrations of BiOCl (insert: Photographs of BiOCl-doped adhesives at Bi concentration of 105 mM). **f** FTIR spectra of BioGlue, Bi-BioGlue, BSA and GA. **g** SEM image and elemental mapping images of Bi-BioGlue. **h** Dynamic oscillatory time sweep measurement of BioGlue and Bi-BioGlue. **i** The pattern of "TMU" formed by Bi-BioGlue and the corresponding CT image. Spectral CT signal curves (**j**) and corresponding phantom imaging (**k**) of Bi-BioGlue and Iohexol-doped BioGlue at 105 mM at different monochromatic energies. The elements are created in BioRender. Sun, S. (2025) https://BioRender.com/y3dwr6u.

operating at high and low tube voltages. Through computer analysis and processing, spectral CT generates images based on the absorption of single-energy X-rays by the scanned object. Spectral CT offers unique capabilities, such as monochromatic imaging, material separation, effective atomic number determination, and removal of metal artifacts[51-55]. Due to the much higher K edge energy value of Bi (91 keV) compared to the I (33.2 keV), the CT intensity of I-BioGlue showed sharply attenuation decay curve and at high monochromatic X-ray energy on spectral CT imaging, while the Bi-BioGlue exhibited a relatively flat decay curve. At 190 keV, the CT value of Bi-BioGlue was 3.9 times higher than that of Iohexol-doped BioGlue. The results demonstrated that Bi-BioGlue exhibited better contrast effects compared to clinical iohexol (Fig. 2j, k).

### Adhesion Ability of Bi-BioGlue

Thereafter, we investigated the adhesion performance of Bi-BioGlue in vitro. The injection apparatus consists of glue gun, bayonet mixing tube and 50 mL double cylinder, which can mix and inject the BiOCl-BSA and GA in a volume ratio of 4:1 (Fig. 3a). Due to the plenty of amine groups in BSA and biological tissues, the GA can interact with them by crosslink reaction to realize adhesion. To evaluate the adhesive performance, BioGlue and Bi-BioGlue were applied in the interlayer of two porcine aortic vessels (about 1 cm²), respectively. The vessels were tightly connected by Bi-BioGlue and BioGlue, with maximum with standable pulling forces of 4.88 N and 4.87 N, respectively, indicating that the adhesive strength between Bi-BioGlue and BioGlue is comparable (Fig. 3b). Then, an ex vivo porcine aorta adhesion test was carried out to further evaluate the adhesive ability of Bi-BioGlue. The aorta's lower end was tied off with sutures, and a gap was created at the aortic bifurcation. When red dye simulated blood flowed into the upper end, and fluid outflow from the gap was observed. After connecting the vessel gap using Bi-BioGlue, the fluid outflow ceased completely, thus confirming the successful closure of the gap (Fig. S10).

### In Vivo Monitoring of Bi-BioGlue by CT Imaging

Next, the metabolic process of Bi-BioGlue was investigated by CT imaging. 30 μL of Bi-BioGlue was surgically adhered to the posterior vena cava of SD rats, and CT imaging revealed significant contrast enhancement at the adhesion site, with a CT value as high as 260 HU. The Bi-BioGlue maintained stable morphology and size within the first week, with only a gradual decrease in CT signal intensity. This indicates that the initial degradation of Bi-BioGlue is very slow and that there is no apparent presence of unlabeled adhesive. As time progressed, Bi-BioGlue exhibited further degradation, as evidenced by a gradual reduction in size on CT imaging, eventually the CT value of adhesion

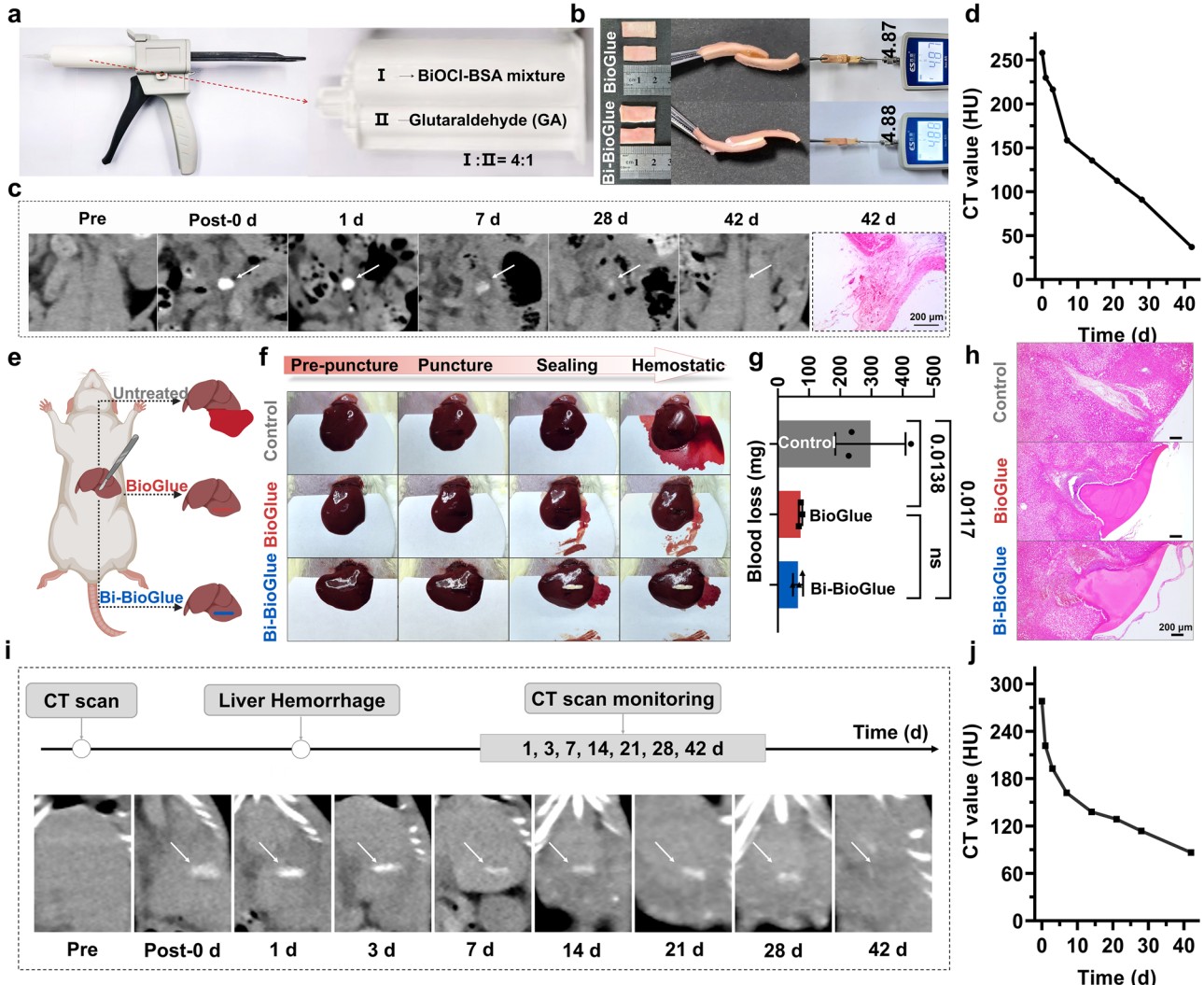

**Fig. 3 | CT imaging of Bi-BioGlue in vivo. a** The photograph of injection apparatus. **b** Porcine aortic vessels adhered with BioGlue or Bi-BioGlue for tension testing with a tensile tester. **c** CT images and HE staining images of adhesive site after surgical adhesion to the posterior vena cava for 42 days. **d** Corresponding CT value change curve of (**c**). **e** Schematic diagram of establishment of liver hemorrhage model and treatment process. **f** Photographs of liver before and after different treatments. **g** The amount of blood loss after different treatments. (*n* = 3 rats; mean ± SD; statistical analysis: one-way ANOVA; ns, no significands; P values are shown in the figure). **h** HE staining images of liver at injury sites at 24 h post-treatment. CT images of liver (**i**) and corresponding CT value change curve (**j**) after treating the hemorrhage with Bi-BioGlue. The elements are created in BioRender. Sun, S. (2025) https://BioRender.com/v99tjw5.

site returning to the soft tissue level (40 HU) by day 42 (Fig. 3c, d). Meanwhile, HE staining confirmed almost complete degradation of Bi-BioGlue in the adhesive region (Fig. 3c). The above results indicated that Bi-BioGlue possessed a relatively prolonged imaging time window and could undergo gradual degradation in vivo. Given the negligible $Bi^{3+}$ release of BiOCl under simulated physiological conditions in vitro, it is reasonable to infer that BiOCl retains considerable labeling efficiency even during later stages of degradation (Fig. S11). Considering that the intraoperative and early postoperative periods represent high-risk windows for embolus detachment when using BioGlue, the demonstrated in vivo labeling efficiency of Bi-BioGlue during the early stage ensures the reliable detection of emboli within this critical timeframe.

**In Vivo Hemostasis and Healing on Rat Liver**
After confirming their adhesive property and in vivo CT imaging capacity, Bi-BioGlue was applied to hemostasis in a liver puncture rat model (Fig. 3e). Hemorrhagic liver injury model was established by surgical creation of a wound 10 mm long and 3 mm deep. Subsequently, BioGlue and Bi-BioGlue were employed to treat the bleeding

wound, and the untreated group set as a control. Both BioGlue and Bi-BioGlue achieved wound closure and hemostasis within 30 s, whereas severe blood loss occurred in the control group without any adhesive application (Fig. 3f). The amount of bleeding in control group, BioGlue group, and Bi-BioGlue group throughout the hepatic hemostasis process was measured to be 300 mg, 70 mg, and 65 mg, respectively. Bi-BioGlue exhibited a hemostatic effect similar to BioGlue, significantly reducing blood loss by 78.4% compared to the control group, thereby demonstrating the excellent hemostatic capability of Bi-BioGlue (Fig. 3g). The attachment of Bi-BioGlue to the liver was also proved by HE analysis after 24 h (Fig. 3h). Furthermore, the location and morphology of Bi-BioGlue were monitored by CT imaging, revealing gradual degradation over 42 days, consistent with pathological finding (Fig. 3i, j).

**In Vitro CT Imaging of Bi-BioGlue**
The detection limit of Bi-BioGlue was investigated for emboli of different sizes embedded within red meat of varying thicknesses (Fig. S12a). Ex vivo CT imaging showed that emboli with diameters of 1.0 mm (edge length of cubic emboli) and 1.2 mm. exhibited CT values

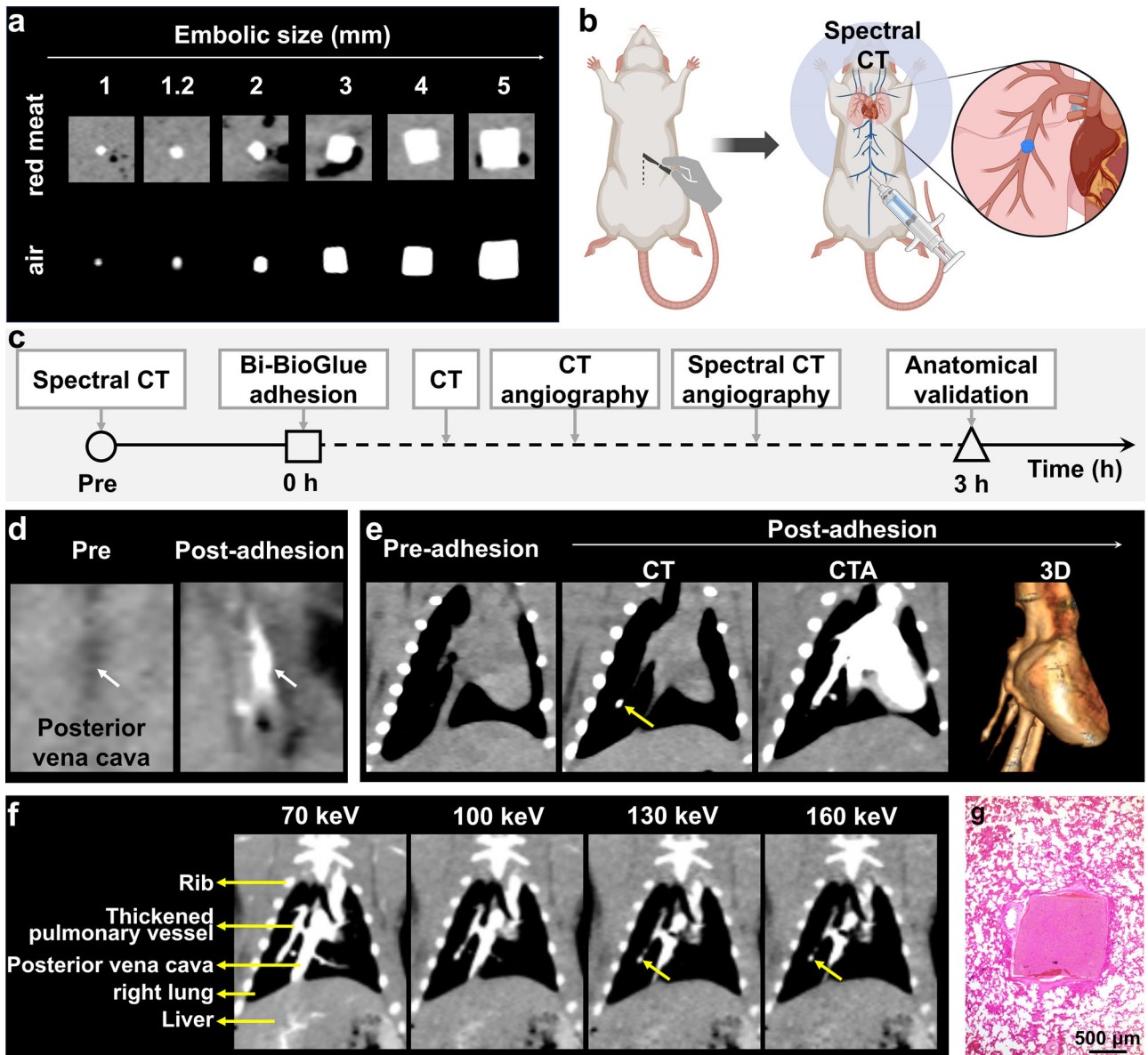

**Fig. 4 | CT imaging of Bi-BioGlue-induced pulmonary embolism. a** CT images of Bi-BioGlue emboli with different sizes in air and in red meat (The images in the first row were representative regions extracted from Fig. S12). **b** Schematic diagram of pulmonary embolism in SD rats caused by Bi-BioGlue embolus. **c** Flow diagram of spectral CT imaging. **d** CT images of posterior vena cava before and after Bi-BioGlue adhesion (adhesive site of Bi-BioGlue was indicated by the white arrow). **e** Spectral CT images of the chest before and after modeling, followed by CTA scanning and corresponding 3D reconstruction (Embolus was indicated by the yellow arrow). **f** CTA images after pulmonary embolism at different monochromatic energies (Bi-BioGlue-induced embolus was indicated by the yellow arrow). **g** Histological analysis of pulmonary embolus. The elements are created in BioRender. Sun, S. (2025) https://BioRender.com/3j2wtpe.

of approximately 300 HU and 500 HU, respectively. These values were slightly lower than those of bulk samples due to partial volume effects. Notably, the CT values of Bi-BioGlue emboli were not significantly affected by the presence of varying thicknesses (1.0, 3.0, 5.0 cm) of overlying red meat, demonstrating the excellent tissue penetration capability of CT imaging.

Furthermore, we explored the influence of CT scanning parameters on embolus detectability, with particular attention to tube voltage and slice thickness. To achieve optimal visualization of ultrasmall emboli, the slice thickness was set to the minimum value supported by the instrument (0.5 mm). When the tube voltage was varied between 80 and 140 kV, only minor fluctuations in CT values were observed across different embolus sizes, which can be attributed to the high K-edge energy of Bi (Fig. S12b, corresponding to Fig. 4a). These results indicated that changes in these scanning

parameters did not significantly affect the embolus detection capability.

**In Vivo CT Imaging of Bi-BioGlue-induced Pulmonary Embolism**
To assess the feasibility of CT visualization for early and precise detection of adhesion emboli formed by surgical adhesives in vivo, an adhesive-induced pulmonary embolism model was established by puncturing and injecting a Bi-BioGlue embolus into the posterior vena cava, followed by sealing the puncture sites with Bi-BioGlue. Once entering the bloodstream, the emboli would flow with blood into pulmonary circulation and rapidly occlude the pulmonary artery within seconds or minutes (Fig. 4b). Then in vivo pulmonary embolism experiment was carried out according to the timeline of Fig. 4c. After modeling, the CT scan showed the location and pattern of Bi-BioGlue at posterior vena cava (Fig. 4d), meanwhile, a small bright nodule was

observed obviously at middle area of right lung. Subsequently, spectral CT angiography (CTA) was performed through intravenous injection of iodine-based contrast agent (Iohexol). The CT images under mixed X ray energies showed a visibly dilated pulmonary artery and abruptly interrupted in the middle area of right lung, indicating the occurrence of right pulmonary embolism, with a bright nodular signal at the occlusion site corresponding to the Bi-BioGlue embolus. (Fig. 4e). Compared to ex vivo conditions, the CT values of 1.0 mm and 1.2 mm Bi-BioGlue emboli decreased to approximately 220 HU and 400 HU, respectively, in vivo (Fig. S13). Owing to its small size and a CT value only slightly higher than that of surrounding tissues, the 1.0 mm embolus was difficult to reliably identify in vivo. In contrast, the 1.2 mm embolus exhibited markedly enhanced contrast against background tissue and was clearly visualized. Therefore, 1.2 mm was determined to be the minimum detectable embolus size in vivo using Bi-BioGlue.

Besides, monochromatic X-ray energy images of Spectral CTA were further constructed to elucidate the relationship between the abnormal nodule and the dilated artery. As Bi element possesses a higher K- edge value (91 keV) compared to iodine (33.2 keV), Bi-BioGlue exhibited strong X-ray attenuation in both low and high monochromatic X-ray energies, whereas Iohexol showed weak X-ray attenuation at high monochromatic X-ray energy (Fig. S14)[56,57]. Therefore, Bi-BioGlue could be distinguished from Iohexol using spectral CT. At low monochromatic X-ray image of 70 keV, both the dilated artery and abnormal nodule displayed high signals and blended. When increasing the monochromatic X-ray energy, the signal of dilated artery was sharply declined and could not even be clearly displayed at 130 and 160 keV, while the nodule was still visible and located at the end of dilated artery, demonstrating the nodule was embolized into the pulmonary artery and resulting in dilation of upstream vessels (Fig. 4f and Fig. S15). Additionally, HE analysis verified the pulmonary embolism induced by Bi-BioGlue (Fig. 4g). These experiments proved that incorporation of BiOCl in adhesive enables early and accurate diagnosis of adhesive embolism complication of BioGlue by CT imaging. Therefore, our approach enhanced BioGlue's safety by addressing the clinical challenge of undetectable embolic complications, rather than altering its chemical composition. By enabling reliable post-operative monitoring of emboli, Bi-BioGlue provided a more comprehensive and proactive safety profile.

### In Vitro Spectral CT Imaging for Differentiation of Bi-BioGlue-induced Embolus from Confusable Calcified Nodule

To further substantiate the clinical utility of Bi-BioGlue, we conducted in-depth study into the detection capability of emboli formed by Bi-BioGlue under complex background interference. In CT imaging, images of target areas are often affected by intracorporeal calcifications, such as vascular calcified plaques or pulmonary calcified nodules[58–60]. Therefore, distinguishing emboli caused by detachment of Bi-BioGlue from these calcifications presents a significant challenge. Considering that calcified plaques and nodules are mainly formed by organic material and calcium salt deposition, we utilized $Ca_3(PO_4)_2$-doped BioGlue to simulate calcification nodules for in vitro and in vivo studies. To assess the specificity of spectral CT imaging in identifying Bi-BioGlue embolus in vitro, I-BioGlue, Bi-BioGlue, BioGlue injected into a well at row E and column 3, 6, 9 of 96-well plate, respectively, while filling the other remaining wells with $Ca_3(PO_4)_2$-doped BioGlue (Fig. 5a). It was not surprisingly that the Bi-BioGlue was difficult to be distinguished from I-BioGlue, Ca-BioGlue in conventional CT (Fig. 5b). CT imaging results showed that Bi-BioGlue consistently maintained a high brightness across the entire range of X-ray energies, whereas BioGlue alone remained undetectable at all energy levels. Both I-BioGlue and Ca-BioGlue exhibited strong brightness at low X-ray energies, but their signals declined sharply as the X-ray energy increased, becoming extremely weak at higher energies (Fig. 5c, d). These findings clearly demonstrated that Bi-BioGlue enabled effective differentiation between emboli and calcified tissues in spectral CT imaging. In spectral CT, both Bi-BioGlue and Ca-BioGlue were clearly visible and indistinguishable from each other at low monochromatic X-ray energies of 40 and 70 keV. With the increase in monochromatic X-ray energy, the brightness of Ca-BioGlue's CT images began to decrease and became almost invisible above 130 keV, whereas Bi-BioGlue maintained a relatively high contrast enhancement and could be clearly identified in the 96-well plate.

### In Vivo Spectral CT Imaging for Differentiation of Bi-BioGlue-induced Embolus from Confusable Calcified Nodule

Subsequently, a rat model of pulmonary adhesive embolization concurrent with a calcified nodule was established by injecting a Bi-BioGlue embolus from the posterior vena cava and introducing a calcified nodule via intercostal space puncture. At low monochromatic X-ray energies of 40 and 70 keV, bone structures (such as ribs), Bi-BioGlue, and calcified nodules all exhibited pronounced bright effects on both sagittal and transverse CT images (Fig. 5e, f). Particularly, Bi-BioGlue emboli and calcified nodules showed similar morphology, signal-to-noise ratio (SNR) and contrast-to-noise ratio (CNR) (SNR: 5.22 vs 5.28; CNR: 73.41 vs 60.92) (Fig. 5g, h). This similarity led to confusion and difficulties in detecting adhesive emboli. The emboli formed by Bi-BioGlue maintained a relatively high contrast enhancement; however, the enhancement effect of calcified nodules sharply declined. At 190 keV, the SNR of Bi-BioGlue was 4.05 times higher than that of the calcified nodule (Fig. 5g), demonstrating the superior contrast of Bi-BioGlue emboli compared to calcified nodules. These results demonstrated that even in the presence of complex background interference, Bi-BioGlue emboli can still be accurately detected using spectral CT imaging.

### Biosafety Evaluation of Bi-BioGlue

The in vitro cytotoxicity was investigated by standard MTT assay. H9c2 cells and HUVEC, two of the most commonly used cell lines in cardiovascular research were cultured in media containing various concentrations of Bi-BioGlue for 24 and 48 h. The cell viabilities of H9c2 cell and HUVEC lines were remained higher than 80% even at a concentration as high as 20 mg/mL Bi-BioGlue which results suggested low cytotoxicity of Bi-BioGlue towards H9c2 cells and HUVEC (Fig. S16a, S16b).

For in vivo toxicology study, the rats were injected with Bi-BioGlue near the posterior vena cava, and then body weight monitoring, biochemical index analysis and hematoxylin and eosin (H&E) staining were carried out. During the 45 days' monitoring, there was no significant body weights change between experimental group and control group (Fig. S16c). At days 1, 15, and 45 post-operation, the biochemical indexes (liver function markers: alanine aminotransferase, ALT; aspartate aminotransferase, AST; kidney function markers: urea, UREA; creatinine, CREA) and hematological parameters (white blood cell, WBC; red blood cell count, RBC; hemoglobin concentration, HGB; platelet count, PLT) of the experimental group rats were comparable to those of the control group (Fig. S16d, S16e). Moreover, the H&E slice images of major organs including heart, liver, spleen, lung and kidney showed no evident lesion, necrosis, and inflammation (Fig. S16f). The systematic toxicological assessment demonstrated Bi-BioGlue did not alter the intrinsic biosafety profile of the original BioGlue in terms of chemical composition.

## Discussion

Surgical adhesives are widely used for hemostasis, reinforcement of suture lines, and repair of fragile tissues, significantly improving the safety and efficacy of surgical procedures. Despite these benefits, leakage or detachment of surgical adhesives may lead to the severe complication of vascular embolism. As the most extensively used surgical adhesive, BioGlue carries a risk of generating adhesive-derived

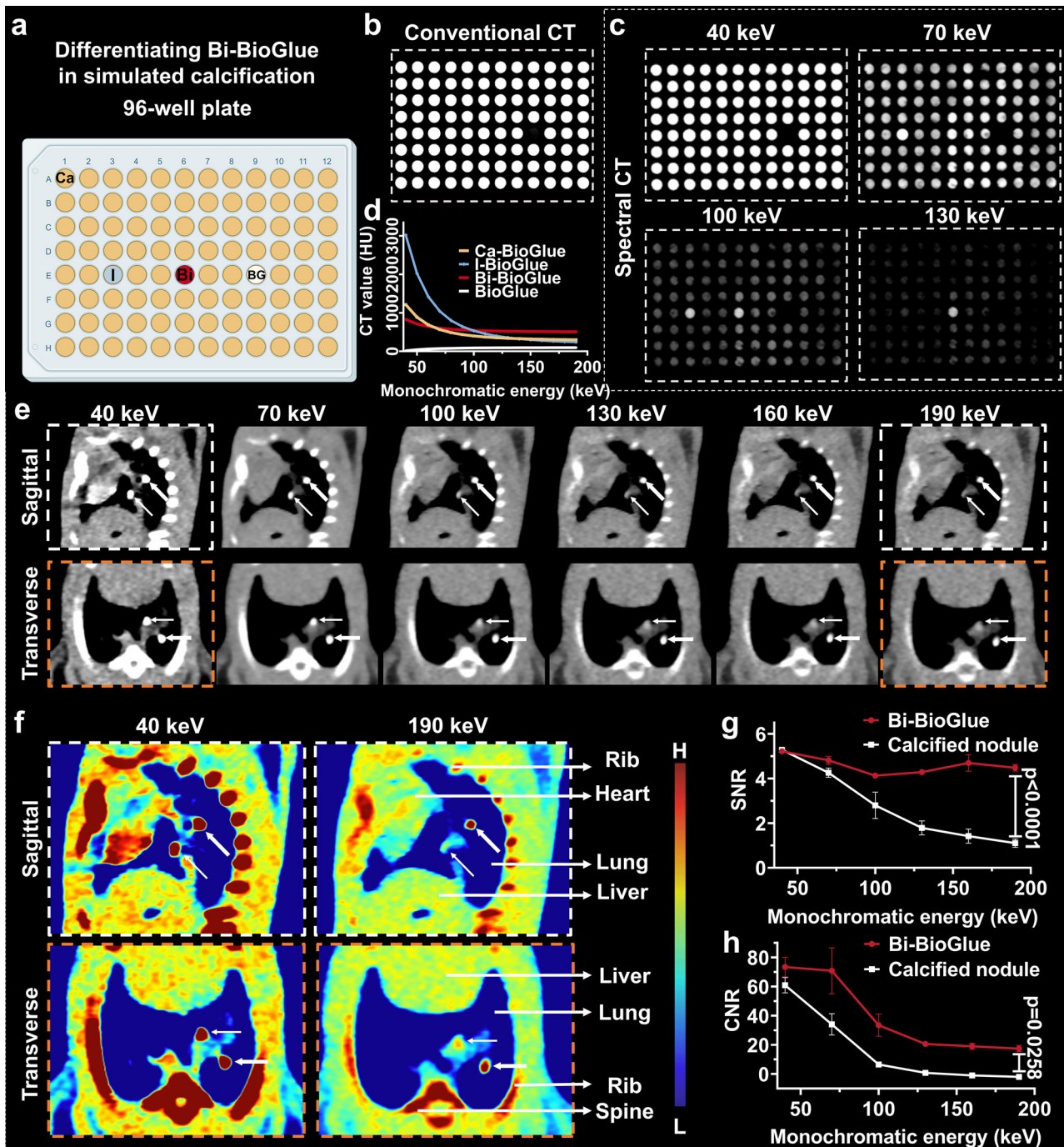

**Fig. 5 | Spectral CT imaging for differentiation of Bi-BioGlue-induced embolus from confusable calcified nodule. a** Schematic illustration of I-BioGlue injected into a well at row E and column 3, Bi-BioGlue injected into a well at row E and column 6, BioGlue injected into a well at row E and column 9 of 96-well plate, while filling other wells with $Ca_3(PO_4)_2$-doped BioGlue. Conventional CT images (**b**) and spectral CT images (**c**) of 96-well plate. **d** Spectral CT value curves of Bi-BioGlue, I-BioGlue, BioGlue and $Ca_3(PO_4)_2$-doped BioGlue at different monochromatic energies. **e** Sagittal and transverse spectral CT images for the pulmonary adhesive emboli during or after surgery, which commonly occur in pulmonary, coronary, and cerebral vessels. Emboli of different sizes may impair tissue perfusion, resulting in ischemia, necrosis, and in severe cases cerebral or myocardial infarction. These emboli are difficult to treat using conventional anticoagulant or thrombolytic therapies and often require interventional or surgical removal. Early and accurate detection is therefore critical.

embolization concurrent with a calcified nodule model at different monochromatic energies. **f** Corresponding pseudo color images of (**e**) at monochromatic energies of 40 and 190 keV. The calcified nodule is indicated by a thin arrow, and the Bi-BioGlue embolus is indicated by a wide arrow. The SNR (**g**) and CNR (**h**) of Bi-BioGlue embolus and simulated calcification nodule (**e**) at different monochromatic energies ($n = 3$ biologically prepared samples; mean ± SD; statistical analysis: two-way ANOVA; P values are shown in the figure). The elements are created in BioRender. Sun, S. (2025) https://BioRender.com/j0ynkjm.

Clinically, CTA and DSA are commonly used to detect vascular embolism by identifying disrupted contrast flow. However, their reliance on contrast injection limits early diagnosis, and these indirect methods cannot directly visualize emboli or determine their origin. Although IVUS and OCT can directly visualize BioGlue-induced emboli, their invasive nature and limited penetration restrict routine clinical use. Consequently, there is a strong need for a

noninvasive and precise method to detect adhesive-induced vascular embolism.

In this study, we demonstrate, for the first time, the hypersensitive in vivo detection of single millimeter-scale vascular emboli derived from surgical adhesives using CT imaging. By systematically comparing Bi-based imaging probes with distinct physicochemical properties, we identify BiOCl as uniquely capable of achieving homogeneous incorporation within BioGlue while maintaining long-term labeling stability. This stable integration endows both the adhesive and its embolic fragments with high CT imaging sensitivity and sustained imaging contrast, distinguishing BiOCl from other Bi-based candidates that fail to meet these criteria. Importantly, this labeling strategy is operationally simple, broadly applicable to different adhesive systems, and does not interfere with the intrinsic gelation process of the adhesive, highlighting its practical feasibility.

Using Bi-BioGlue as a representative model, we further investigate the imaging performance of this strategy across multiple biologically relevant scenarios, including wound adhesion, in vivo localization, degradation tracking, and emboli detection. CT imaging enabled continuous monitoring of the spatial distribution and in vivo fate of Bi-BioGlue over a 42-day period, during which gradual material degradation was clearly observed. These observations indicate that the proposed approach is suitable for long-term in vivo surveillance of adhesive behavior. In a rat pulmonary embolism model, adhesive emboli as small as 1.2 mm were reliably detected with high sensitivity. Notably, spectral CT imaging leveraging the K-edge characteristics of Bi enabled accurate discrimination of Bi-BioGlue emboli even in the presence of confounding lung calcifications, a condition that often limits the specificity of conventional CT-based diagnosis. In parallel, systematic in vitro and in vivo toxicological evaluations confirmed the favorable biocompatibility of Bi-BioGlue, supporting its suitability for biomedical applications.

Despite these advances, several limitations of the present study should be acknowledged. CT imaging of Bi-BioGlue emboli was primarily evaluated in pulmonary embolism models, and validation in other clinically relevant settings, such as coronary, carotid, or cerebral embolism, was not performed. In addition, the current investigations were restricted to rat models, and the emboli examined were pre-fabricated rather than spontaneously generated during adhesive application. At this stage, biosafety evidence is derived solely from short-term rat studies, with long-term toxicity and validation in large-animal models yet to be conducted. Future studies will therefore focus on clinically relevant adhesive procedures in large animals, such as pigs, to better recapitulate the natural formation of adhesive emboli and to evaluate detection performance under conditions relevant to clinical translation. Extended longitudinal biosafety studies will also be undertaken to further substantiate the clinical potential of Bi-BioGlue. In parallel, integration with advanced photon-counting CT may further expand the capabilities of this approach for element-specific identification and detection of Bi-BioGlue emboli. Collectively, this work provides a practical framework for the early and precise detection of adhesive-related emboli and offers a pathway toward improved biosafety in the expanding clinical use of surgical adhesives.

## Methods

### Materials
All chemicals used were of at least analytical grade. BSA, GA, $Bi_2S_3$, $Bi_2O_3$, $BiCl_3$ and methyl thiazolyl tetrazolium (MTT) were purchased from Shanghai Aladdin Biochemical Technology Co., Ltd. (Shanghai, China). Dimethyl sulfoxide (DMSO) and ethanol were purchased from Concord Technology (Tianjin, China). Polyformaldehyde was obtained from Service Bio-Biotechnology Co., Ltd. (Wuhan, China). Ultra-pure water (Hangzhou Wahaha Group Co., Ltd.) was used throughout the synthesis process.

### Synthesis of BiOCl and Bi-DTPA
In a typical produce, $BiCl_3$ (1.5 mmol) was dissolved in 10 mL of deionized water, followed by ultrasonication for 10 min. The products were purified by centrifugation ($424 \times g$, 2 min) with ultrapure water three times.

The Bi-DTPA was fabricated according to our group reported method[46]. Briefly, $Bi_2O_3$ (0.5 mmol) and DTPA (1 mmol) were mixed in 20 mL $H_2O$ and stirred vigorously at 85 °C for 2 h. The dark yellow suspension gradually became clear, indicating Bi-DTPA formation (pH=1.5). The pH was then adjusted to 7.4 with NaOH, and the Bi-DTPA product was freeze-dried for further use.

### Synthesis of BioGlue, Bi-based BioGlue and I-doped BioGlue
The BioGlue used in this study was synthesized according to commercialized BioGlue product, which is a medical surgical adhesive developed and marketed by the American company CryoLife in the 1990s. It is a two-component adhesive, consisting of a 45% BSA solution and a 10% glutaraldehyde solution mixed in a 4:1 volume ratio.

To synthesize of Bi-based BioGlue, BiOCl, Bi-DTPA, $Bi_2S_3$ or $Bi_2O_3$ was first mixed and stirred with BSA solution to get Bi probe/BSA mixture with concentrations of Bi element at 22–131 mM and BSA at 450 mg/mL, followed by the addition of glutaraldehyde in a volume ratio of 4:1 to form Bi-based BioGlue. I-doped BioGlue was synthesized through replacing the Bi probe with Iohexol (22–131 mM) and other steps were the same as those of Bi-based BioGlue synthesis.

### Synthesis of Hf-, Ta-, Yb-doped BioGlue
To synthesize Hf-, Ta-, and Yb-doped BioGlue, $HfCl_4$, $TaCl_5$, and $YbCl_3$ were individually mixed and stirred with BSA solution to obtain metal probe/BSA mixtures containing 131 mM of the metal element and 450 mg/mL of BSA, respectively. Glutaraldehyde was then added at a volume ratio of 4:1 to form the corresponding Hf-, Ta-, and Yb-doped BioGlue.

### Characterization
Transmission electron microscopy (TEM) (HT7700, Japan) was employed to characterize the morphology of BiOCl and Bi-DTPA. The morphology of Bi-BioGlue was characterized using FEI scanning electron microscopy (SEM) (Axio-Imager_LSM-800, German). Fourier transform infrared spectra (400-4000 $cm^{-1}$) of BSA, GA, and Bi-BioGlue adhesives were recorded with a NicoletiS10 spectrometer (Nicolet, USA), utilizing pure KBr as the background material. The X-ray diffraction (XRD) analysis was conducted using an UltimaIV X-ray diffractometer (Rigaku, Japan). Rheology experiments of BioGlue, Bi-, Hf-, Ta-, Yb-BioGlue were tested on an MCR xx2 rheometer (Anton Paar, Austria).

### Stability Evaluation
To evaluate the morphological stability of Bi-BioGlue, a Bi-BioGlue cube with size of $5 \times 5 \times 5$ mm was immersed in different media, including phosphate buffer solution (PBS, pH = 7.4), normal saline, and Dulbecco's modified eagle medium (DMEM, Gibco, USA). The stability of these samples was monitored by taking photos daily for 14 days.

To study the CT imaging stability of Bi-BioGlue, a Bi-BioGlue cube measuring $5 \times 5 \times 5$ mm was immersed in ultrapure water, and changes in CT values were monitored via CT scans at various time intervals (1, 2, 4, 8, 12, and 24 h). For comparison, Bi-DTPA-BioGlue was also evaluated under the same conditions as those for Bi-BioGlue.

Both the Bi-BioGlue cube ($5 \times 5 \times 5$ mm) and BiOCl with equivalent Bi content were incubated in four representative media: $H_2O$, physiological buffer (10 mM PBS, pH 7.4), an oxidative stress environment (1 mM $H_2O_2$), and serum. After different incubation periods (1, 7, and 14 days), the released Bi ions were quantified using Inductively Coupled Plasma Optical Emission Spectrometry (ICP-OES) analysis. Subsequently, the samples were freeze-dried for XRD characterization

## Tensile Test

The tensile test is a widely used experimental method for evaluating the strength and adhesive properties of materials. Two clean porcine aortic blood vessels (each measuring $1 \times 2$ cm) were bonded using either Bi-BioGlue or BioGlue, with a bonded area of $1$ cm². After bonding, tensile tests were performed utilizing a mechanical tester to measure the force applied until the bonded vessels ruptured. The maximum pulling force was recorded to assess the adhesive capability of Bi-BioGlue.

## Ex Vivo Tissue Adhesion of Bi-BioGlue

To evaluate the tissue adhesive performance of Bi-BioGlue, an ex vivo leakage model was established using isolated pig aorta vessels. A longitudinal incision was made on the vessel wall to simulate a leakage site. Bi-BioGlue was used to the treatment of the defect using a syringe, followed by 30 s of contact to initiate adhesion. The repaired vessel was then allowed to stand undisturbed for 120 s to ensure complete gelation and bonding. Subsequently, red ink was injected from the proximal end of the vessel to simulate blood flow. The absence of ink leakage from the repair site was used as an indicator of effective tissue adhesion.

## Cytotoxicity

Rat cardiomyocytes (H9c2) cells and HUVEC were purchased from Cell Bank of Chinese Academy of Sciences and used to investigate the cytocompatibility of Bi-BioGlue in vitro. Firstly, the freeze-dried Bi-BioGlue were soaked in the culture medium with different concentrations (0, 1, 3, 5, 10, 15, 20 mg/mL) and incubated for 24 h at 37 °C to obtain the Bi-BioGlue extracts. H9c2 cells and HUVEC were cultured in 96-well plates with a density of $1 \times 10^4$ per well for 24 h at 37 °C with 5% $CO_2$. Afterwards, the cells were rinsed with phosphate buffer solution (PBS, 10 mM, pH = 7.4) and Bi-BioGlue extractswas added to the wells and incubated for another 24 h or 48 h. Subsequently, H9c2 cells and HUVEC were rinsed with phosphate buffer solution and fresh culture medium supplemented with MTT (5 mg/mL, 10 μL) was added to the cells and incubated for 4 h at 37 °C with 5% $CO_2$. The MTT-containing medium was then replaced, and DMSO (120 μL) was added to dissolve the blue-purple formazan crystals formed at the bottom of the wells. After 10 min of gentle shaking, 490 nm absorption value of each well were measured using a microwell plate reader (Bio-Tek, USA). The cell viability was defined and calculated as follows: Cell viability $= OD_{exp}/OD_{con} \times 100\%$, of which $OD_{exp}$ is the optical density (OD) of cells cocultured with diverse concentrations of Bi-BioGlue adhesive, and $OD_{con}$ means that of the control group.

## In Vivo Toxicity Study

All animal experiments were performed in compliance with the guidelines of the Animal Care and Use Committee of Tianjin Medical University General Hospital, and approved by the Animal Care and Use Committee of Tianjin Medical University General Hospital (IRB2022-DW-76). Sprague-Dawley (SD) rats (280-320 g) were provided by SPF Biotechnology Co., Ltd. (Beijing, China). Animals were bred and housed under a 12:12 h light-dark cycle at an ambient temperature of 22 °C and a relative humidity ranging from 40 to 70%.

Sixteen male SD rats were randomly divided into four groups (Control group, 1-, 15- and 45-day groups), with four rats in each group. Rats in 1-, 15- and 45-day groups were anesthetized with isoflurane and underwent a midline abdominal incision to expose the posterior vena cava for injection of 30 μL of Bi-BioGlue, followed by suturing the incisions. Rats in control group underwent a similar surgical procedure without receiving Bi-BioGlue. The body weight changes of rats in control group and 45-day group were recorded every three days. After 1, 15, and 45 days post-operation, the rats in the corresponding group were sacrificed, and the rats in Control group were sacrificed after 45 days post-operation. Major organs including heart, liver, spleen,

lung and kidney were fixed in paraformaldehyde for histopathological analysis. Blood samples were collected for biochemical analysis and routine blood tests.

## In Vivo Monitoring of Bi-BioGlue via CT Imaging

SD rats ($n = 3$) were anesthetized with isoflurane and placed in supine position for abdominal fur removal. A 2 cm incision was made along the midline of the abdomen to expose the posterior vena cava. Bi-BioGlue was injected to the posterior vena cava, followed by free the clips and suturing the abdominal incision. After that, spectral CT scanning was performed on a clinical CT scanner (Somatom Drive, Siemens Healthineers, Erlangen, Germany) to monitor the changes of Bi-BioGlue colloid at 1, 7, 28 and 42 days post-operation under following scanning parameters: field of view (FOV) was $256 \times 256$ mm, slice thickness was 0.5 mm, with adaptive tube current and voltage were set at 80/140 Sn.

## In Vivo Hemostasis and Healing on Rat Livers

SD rats were divided into three groups ($n = 3$ for each group), including control group, Bi-BioGlue group and BioGlue group. The rats ($n = 3$ for each group) were anesthetized with isoflurane and placed in supine position for abdominal fur removal. A 10 mm lateral incision was made to expose the liver, followed by placing a filter paper under the liver. A 10 mm long, 3 mm deep incision was made on the left hepatic lobe with a surgical scalpel. After inducing liver bleeding, rats in the Bi-BioGlue and BioGlue groups were treated with 60 μL of Bi-BioGlue or BioGlue, respectively. Both adhesives sealed the bleeding site and stopped hemorrhage within approximately 30 s. Rats in the control group did not receive any treatment and were allowed to bleed and coagulate naturally. The liver was returned to its original position in the abdomen and the incision was sutured. The filter paper was weighted before and after operation to quantify the amount of blood loss. Spectral CT scan was carried out before and after surgery (1, 3, 7, 14, 21, 28, and 42 days post-operation). scanning parameters: FOV was $256 \times 256$ mm, slice thickness was 0.5 mm, with adaptive tube current and voltage were set at 80/140 Sn.

SD rats were divided into three groups ($n = 3$ for each group) randomly and treated as mentioned above. At 24 h post-operation, the rats were euthanized, and the liver were harvested for histopathological analysis of operation site.

## Ex Vivo CT imaging of Bi-BioGlue Emboli with Different Parameters

Bi-BioGlue was carefully trimmed into cubic emboli with diameters of 1.0, 1.2, 2.0, 3.0, 4.0, and 5.0 mm. CT imaging was conducted under various conditions to evaluate the detectability of emboli in ex vivo settings.

Scanning parameters optimization: emboli were imaged while embedded in red meat. Slice thicknesses were varied at 0.5, 1.0, 2.0, 3.0, 4.0, 5.0, and 10.0 mm, and the tube voltage was adjusted between 80, 100, 120, and 140 kV.

Impact of penetration depth on the detectability of Bi-BioGlue emboli: To assess the penetration capability of CT, 1.2 mm emboli were sandwiched between red meat slices of 1.0, 3.0, and 5.0 cm thickness. CT scanning was conducted with a FOV of $256 \times 256$ mm, slice thickness of 0.5 mm, and tube voltage of 120 kV.

Detection limits in air and tissue: emboli were placed either in air or sandwiched between two 2 cm-thick slices of red meat. Scanning was performed with a FOV of $256 \times 256$ mm, a slice thickness of 0.5 mm, and adaptive tube current at 120 kV.

## In Vivo CT Visualization of Bi-BioGlue-induced Pulmonary Embolism

SD rats ($n = 3$) were anesthetized with isoflurane and placed in a supine position for abdominal fur removal. A 2 cm incision was made along

the median line of the abdomen to expose the inferior vena cava. The lower end of posterior vena cava, bilateral renal vein and common iliac veins were clamped with vascular clips. A cubic Bi-BioGlue colloid with size of 1.2 mm was injected into the posterior vena cava through a puncture needle. The upper end of posterior vena cava was clamped with vascular clip. After withdrawal of the needle, the puncture site was adhered with Bi-BioGlue, followed by free all clips and suturing the abdominal incision. Before and after operation, spectral CT scanning was performed to visualize the Bi-BioGlue and search for embolus under following scanning parameters: the FOV was set to $256 \times 256$ mm with a slice thickness of 0.5 mm. The adaptive tube current and voltage were 80/140 Sn. SNR and CNR were calculated according to following formula: SNR = CT value of ROI/SD of background, CNR = (CT value of ROI - CT value of muscle)/SD of background. Subsequently, Iohexol enhanced spectral CT was performed to verify the occurrence of embolism under following scanning parameters: the FOV was set to $256 \times 256$ mm with a slice thickness of 0.5 mm. The adaptive tube current and voltage were 80/140 Sn. Spectral images at various monochromatic energies (70-190 keV) were processed using the syngo.via viewer (Siemens Healthineers, Germany). Finally, the rats were euthanized, and histopathological analysis was conducted on the excised pulmonary lobes containing the embolus.

### In Vitro Differentiation of Bi-BioGlue and Calcified Nodule through CT Imaging

To prepare $Ca_3(PO_4)_2$-doped BioGlue, 4.5 g of $Ca_3(PO_4)_2$ and 4 g of BSA are weighed and dissolved in water to form $Ca_3(PO_4)_2$/BSA mixture (final volume 10 mL, $Ca_3(PO_4)_2$ 1.45 mol/L, BSA 400 mg/mL), followed by stirring evenly. It is worth noting that during the preparation of $Ca_3(PO_4)_2$-doped BioGlue, the concentration of BSA used was 400 mg/mL rather than 450 mg/mL. This adjustment was made because a higher BSA concentration, combined with calcium phosphate, is difficult to uniformly disperse in water. Since the CT contrast primarily arises from the calcium content, the slight reduction in BSA concentration is not expected to affect the experimental outcomes. Next, $Ca_3(PO_4)_2$/BSA-doped BioGlue was formed by reacting a mixture of $Ca_3(PO_4)_2$/BSA and 10% glutaraldehyde in a 96-well plate, with volumes of 160 μL/well and 40 μL/well, respectively. In parallel, 200 μL of I-BioGlue, Bi-BioGlue, and BioGlue were added to wells in column E, rows 3, 6, and 9, respectively. The concentrations of iodine and bismuth were adjusted to 105 mM in I-BioGlue and Bi-BioGlue, respectively. The remaining wells were filled with Ca.BioGlue. Subsequently, both conventional CT scanning and spectral CT scanning were carried out following scanning parameters: the FOV was set to $256 \times 256$ mm with a slice thickness of 0.5 mm. The adaptive tube current and voltage were 80/140 Sn, equivalent to a conventional 120 kV. Spectral images at various monochromatic X-ray energies (40-130 keV) were processed using the syngo.via viewer (Siemens Healthineers, Germany).

### In Vivo Differentiation of Bi-BioGlue-induced Pulmonary Embolism and Calcified Nodule through CT Imaging

In this experiment, $Ca_3(PO_4)_2$ doped BioGlue (Ca concentration of 1.16 M) were utilized to simulate calcified nodule. SD rats ($n = 3$) were anesthetized with isoflurane and placed in a supine position for chest and abdominal fur removal. A 2 cm incision was made along the median line of the abdomen to expose the inferior vena cava. The lower end of the renal vein and inferior vena cava, and the upper end of the left and right common iliac veins were clamped with vascular clips. Bi-BioGlue embolus with size of 1.2 mm was injected into the posterior vena cava through a puncture needle to cause pulmonary embolism. After withdrawal of the needle, the puncture site was adhered with Bi-BioGlue, followed by free the clips and suturing the abdominal incision. Then, calcified nodule stimulated with $Ca_3(PO_4)_2$ doped BioGlue was punctured into the lung. After that, spectral CT scanning was

performed to classify of calcified nodule and Bi-BioGlue-induced embolus. Scanning parameters: the FOV was set to $256 \times 256$ mm with a slice thickness of 0.5 mm. The adaptive tube current and voltage were 80/140 Sn, equivalent to a conventional 120 kV. Spectral images at various monochromatic X-ray energies (40-190 keV) were processed using the syngo.via viewer (Siemens Healthineers, Germany). SNR and CNR were calculated according to following formula: SNR = CT value of ROI/SD of background, CNR = (CT value of ROI - CT value of muscle)/SD of background.

### Statistics and Reproducibility

The results in all the experiments were presented as mean ± standard deviation (s.d.). Statistical calculation of experimental data was analyzed using One-Way Analysis of Variance (ANOVA) and Two-Way ANOVA. Article drawing was performed using Origin 2019 or GraphPad Prism 8.0.2 software.

### Reporting summary

Further information on research design is available in the Nature Portfolio Reporting Summary linked to this article.

## Data availability

The data supporting the results in this study are available within the paper and its Supplementary Information. Source data are provided with this paper.

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

## Acknowledgements

Ruihan Liu, Shuo Li and Xingyu Gao contributed equally to this work. This work was supported by the National Natural Science Foundation of China (82071982 (S.-K.S.), 82202830 (C.Z.), 82272052 (J.P.)), Natural Science Foundation of Tianjin City (19JCJQJC63700 (S.-K.S.), 23JCQNJC00620 (C.Z.)). Figure 1 and cartoon images in Figs. 2–5 were created with Biorender.com.

## Author contributions

R.L., Q.Z., X.C., and S.-K.S. conceived and designed the experiments. R.L., S.L., and X.G. performed the experiments. R.L., S.L., X.G., G.S., C.Z., and J.P. participated in data analysis. The manuscript was written by R.L., Q.Z., X.C., and S.-K.S. All authors discussed the results and commented on the paper.

## Competing interests

The authors declare no competing interests.
