## [Transparent Peer Review file · Nature Communications]

Hypersensitive Detection of Single Millimeter Vascular Emboli from Adhesive in vivo

Corresponding Author: Professor Shao-Kai Sun

Version 0:

Reviewer comments:

Reviewer #1

(Remarks to the Author)

The authors incorporated bismuth (Bi) element in the form of BiOCl into a commercial surgical adhesive (BioGlue) for noninvasive detection of adhesive-induced emboli using spectral CT imaging, with detection sensitivity reportedly as small as 1.2 mm. However, despite these findings, the use of bismuth as a CT contrast agent has been extensively documented, and BioGlue, as a commercial surgical adhesive, does not present significant novelty. More importantly, the work lacks sufficient evidence to fully support its claims and conclusions. Therefore, I do not recommend this manuscript for further consideration in Nature Communications. The manuscript can be strengthened by addressing the following points:

1. The authors mention that the CT value of Bi-DTPA-BioGlue decreased within 5 hours when soaked in water, attributing this to the small molecular size and good water solubility of Bi-DTPA. However, how does BiOCl compare? Does BiOCl have a sufficiently large molecular size or poor water solubility to ensure its retention in BioGlue? Clarification on this point is needed.
2. The authors conclude that the CT imaging detection limit for Bi-BioGlue emboli is 1.2 mm. However, this conclusion raises several concerns: (1) In Figure 4a, the 1 mm embolus in red meat is still visible and distinguishable from the surrounding background, calling the 1.2 mm detection limit into question. (2) The authors should verify whether factors such as the amount of Bi doped in BioGlue, the depth of the Bi-BioGlue within the tissue, and the CT machine parameters could influence the detection limit. (3) Additionally, the authors should consider employing a more standardized and scientific method to confirm the detection limit, rather than relying solely on visual judgment.
3. The authors used spectral CT to differentiate Bi-BioGlue from Ca-BioGlue. However, two additional control groups should be included for comparison: (1) BioGlue and Ca-BioGlue, and (2) Iodine-BioGlue and Ca-BioGlue. These controls would better demonstrate that only the developed Bi-BioGlue can effectively distinguish emboli from calcified tissues.
4. In Figures 5e and 5f, the authors should clarify what the two types of arrows represent.
5. The flow of results is disjointed. The authors first discuss Figure 4a, then move to Figure 5a-d, before returning to Figure 4b-g, and finally return to Figure 5e-h. This organization is confusing and should be reconsidered for better coherence.
6. The authors should provide more information on how long it takes for BioGlue to form a pulmonary embolism in vivo.
7. The rationale for using H9C2 cells, a cardiomyocyte cell line, for the biosafety evaluation should be explained, as this cell line may not be the most appropriate model for assessing BioGlue's general biocompatibility.

Reviewer #2

(Remarks to the Author)

The manuscript describes the characterization of a bismuth labelled derivative of a clinically used surgical adhesive, and preclinical in vivo assessment of the feasibility to visualize vascular emboli using this compound using CT imaging. The authors show that the bismuth labelled derivative has the same adhesive capabilities compared to the FDA approved BioGlue.

Major issues that need to be addressed to improve the value of the manuscript:

- In the introduction the size of emboli when damage occurs needs to be mentioned and related to the size detected in the results. The authors also need to explain the benefit of hypersensitive detection of millimetre sized emboli. How do the

authors envision clinical application? Through watch and wait or intervention therapy?

The extent of symptoms will be dependent on the size of the embolism. Will there be differences in treatment compared to standard embolisms? As anticoagulants will not work and nor will fibrinolytic therapy.

- Line 48: Optical imaging is not a suitable imaging methodology for this application due to its limited penetration depth and should be excluded here.

- Line 53: The authors present the bismuth labelling as novel, but this has already been applied for adhesives. Tian et al (Advanced materials 2023) describe a bismuth-pectin organic-inorganic complex that was engineered into a transformable microgel (that they call bi-glue) and that is compatible with x-ray imaging. The statement made by the authors needs to be revised and the paper by Tian et al should be accurately referenced and the differences in approaches should be discussed.

- Fig 3 shows the visualization of emboli over the course of 42 days. What was the percentage of resorbance of emboli? If all were resorbed, what is then the additional value of early detection?

- An explanation should be provided to how detection of millimetre sized emboli will significantly improve the biosafety of surgical adhesions. In my opinion these two features are not directly linked as the biosafety profile of the adhesive is dependent on the chemical composition itself and not on its detectability. Figure 6 shows biosafety assessment of B-BioGlue without comparison to the original BioGlue, therefore significant improvement cannot be claimed.

- Stability assessment of the created complex is insufficient. How can the authors ensure that the complex will remain stable in vivo? This should at least be tested under comparable conditions.

How is the apparent degradation of the adhesive being taken into account? It needs to be investigated that the Bi remains complexed during degradation. The presence of non-labelled adhesive and a possible effect on emboli formation and visualization needs to be assessed.

Reviewer #3

(Remarks to the Author)

The authors present a bismuth-based BiOCl-doped BioGlue for detecting vascular emboli originating from surgical adhesives using CT imaging. This CT-visible adhesive enables in vivo tracking of adhesive degradation and the real-time detection of small emboli. The Bi-BioGlue demonstrates enhanced stability, biocompatibility, and prolonged imaging persistence, with the capability to detect emboli as small as 1.2 mm in a pulmonary embolism model. Overall, this study is well-designed with a rigorous methodology. However, several questions and suggestions need to be addressed before considering publication.

1. The authors used known molecules to form a crosslinked glue. However, the chemistry should be shown more clearly, including the chemical structures of each molecule and the detailed characterizations after synthesis.
2. The cross-linking mechanism between BiOCl and the BioGlue matrix is briefly mentioned, but more experimental details are required. This should include the kinetics and degree of cross-linking, as these factors could directly influence adhesion strength and stability. Additionally, as BiOCl stability is crucial for sustained imaging, please discuss the stability of BiOCl in the adhesive under physiological pH and oxidative conditions.
3. On page 6, line 105, please include a reference for the synthesis method of Bi-DTPA.
4. In Fig S2, please add or label BioGlue, gelatin, and sodium alginate within the figures.
5. In Fig. 2g, please clarify what the colors represent.
6. In Fig. 2k, what are the concentrations of Bi-BioGlue and iohexol-doped BioGlue used in this study? If the concentration is 105 mM, please explain the rationale for this choice. Additionally, please provide a more detailed comparison of detection sensitivity with standard iodine-based agents.
7. In Fig. S3, the color change in DMEM from D0 to D14 (red to yellow) likely reflects pH changes. Please provide an explanation of any chemical reactions or degradation processes that may have occurred during the time period.
8. On page 7, line 141, verify if Fig. 1c is correctly referenced.
9. In Fig. S6, please provide experimental details, including bonding duration and the time elapsed before testing Bi-BioGlue.
10. In Fig. 3f, indicate the time required to achieve sealing and hemostatic effects. This information should be included in the experimental section or figure legend.
11. In Fig. 4a, provide more details regarding the preparation of Bi-BioGlue emboli in air and in red meat.
12. In Fig. 5a, describe the process for creating Ca₃(PO₄)₂-doped BioGlue.
13. On page 15, line 317, ensure consistency in units, as "1.2 mm" appears in several places. Please clarify if this measurement refers to diameter or volume.
14. On pages 9 and 12, where descriptions for Fig. 4 and Fig. 5 are located, reorganize the paragraphs to improve flow and readability.

Reviewer #4

(Remarks to the Author)

Version 1:

Reviewer comments:

Reviewer #2

(Remarks to the Author)

The authors provided additional experimental results to address part of the comments provided by the reviewers. However, the additional data did not cover all issues raised and a number of comments (for all three reviewers) were not or not accurately addressed. Unfortunately, previous questions raised on the novelty of the approach and the degree of translational advance have not been resolved.

The authors are also encouraged to reply to the provided comments only and in a short and concise manner, and exclude lengthy text sections that are not incorporated in the manuscript.

Reviewer #3

(Remarks to the Author)

The authors have made substantial efforts to improve this manuscript, and I believe the revised version is now suitable for publication in its current form.

Reviewer #5

(Remarks to the Author)

This manuscript reported CT imaging-based hypersensitive detection of single millimeter vascular emboli from adhesive in vivo. The proposed Bi-BioGlue with good CT imaging capability can monitor the location, morphology, and degradation process of Bi-BioGlue by CT imaging for a long time, and pulmonary emboli caused by Bi-BioGlue, with size as small as 1.2 mm, can be successfully detected. Additionally, the Bi-Biogule enabled precise detection of pulmonary emboli in spectral CT imaging, unaffected by confounding calcifications. Overall, this study provides a promising non-invasive detection strategy for adhesive emboli.

After the first round of revision, the reviewers' comments were fully addressed, and the quality and level of the manuscript were significantly enhanced. I would like to recommend it for publication in Nature Communications after minor revisions.

Minor issues:

1. In Fig. 2f, the FTIR spectrum of BioGlue needs to be supplemented to further illustrate the impact of the introduction of BiClO on BioGlue.
2. The color of the wells in the 96-well plate in Fig. 5a that contain I, Bi, and BG should use colors with greater distinction, corresponding to the colors in Fig. 5d, which would make it easier to understand.
3. This study achieved good results in animals. However, if it is to be applied in clinical practice in the future, many challenges will still need to be overcome. I suggest that the authors summarize the limitations of this study and the future directions.
4. There are some inconsistencies in formatting and grammatical errors, for instance, both "Bioglu" and "BioGlue" are used. The authors should carefully review the manuscript throughout and make necessary revisions.

Reviewer #6

(Remarks to the Author)

The authors incorporated Bi element in commercial surgical adhesives (BioGlue) for noninvasive detection of adhesive-induced emboli using spectral CT imaging. The results clearly demonstrated that the detecting time is over than 40 days, and the size could limit to 1.2 mm. The authors have carefully revised all the comments. So, in my opinion, the revision should be published as it is now.

Response to Reviewers' Comments

The comments and suggestions made by the reviewers are very helpful for us to revise the manuscript, and we highly appreciate the reviewers for such constructive comments. Detailed reply to the questions and concerns is made below:

Response to Reviewer 1

Comments:

The authors incorporated bismuth (Bi) element in the form of BiOCl into a commercial surgical adhesive (BioGlue) for noninvasive detection of adhesive-induced emboli using spectral CT imaging, with detection sensitivity reportedly as small as 1.2 mm. However, despite these findings, the use of bismuth as a CT contrast agent has been extensively documented, and BioGlue, as a commercial surgical adhesive, does not present significant novelty. More importantly, the work lacks sufficient evidence to fully support its claims and conclusions. Therefore, I do not recommend this manuscript for further consideration in Nature Communications. The manuscript can be strengthened by addressing the following points:

Reply:

Thank you very much for the careful review and valuable comments on our manuscript. We respectfully believe that there may be a misunderstanding regarding the objectives and contributions of our work, and we would like to take this opportunity to clarify these aspects.

First, regarding the issue of novelty, you noted that Bi has been widely used for CT imaging and that BioGlue is a commercially available surgical adhesive. However, the novelty of our work does not lie in the development of a new adhesive, but rather in establishing a sensitive and practical CT imaging strategy to detect emboli formed by the detachment of BioGlue, an unmet clinical need. While Bi-based probes have been explored for imaging applications, their integration with adhesive systems for precise embolus detection requires careful material design and optimization. In our study, we systematically screened a range of Bi-based candidates and identified amorphous BiOCl as a uniquely suitable imaging probe for BioGlue labeling. This represents a novel and rationally engineered solution tailored to the clinical challenge of embolus visualization.

Our work, therefore, introduces a previously unexplored CT imaging approach for tracking BioGlue-derived emboli in vivo with high precision and sensitivity. We believe this contribution offers not only materials innovation and methodological advances, but also substantial clinical relevance.

Second, in response to the concerns regarding data support, we would like to emphasize that the study is grounded in a comprehensive and rigorous experimental framework. This includes rational probe design, detailed physicochemical

characterization, and well-validated in vivo imaging studies. As acknowledged by Reviewer 3, “Overall, this study is well-designed with a rigorous methodology”. We have also conducted additional experiments to further strengthen the conclusions and have revised the manuscript accordingly to address all technical comments. These experiments cover the entire workflow, including material synthesis and characterization, as well as biological evaluation. Specifically, they include solubility and microstructure analysis of BiOCl, assessment of Bi-BioGlue’s stability under physiological conditions, detection limit for emboli, spectral CT imaging performance, and comprehensive biosafety evaluation.

We sincerely thank you again for the rigorous review and constructive suggestions. In the revised manuscript, we have carefully addressed all concerns, and the combination of original and newly added data now provides strong and comprehensive support for our conclusions. We respectfully invite you to re-evaluate our work in light of its demonstrated innovation and the improved robustness of the supporting evidence.

Comment 1:

The authors mention that the CT value of Bi-DTPA-BioGlue decreased within 5 hours when soaked in water, attributing this to the small molecular size and good water solubility of Bi-DTPA. However, how does BiOCl compare? Does BiOCl have a sufficiently large molecular size or poor water solubility to ensure its retention in BioGlue? Clarification on this point is needed.

Reply and corresponding changes:

Thank you very much for your valuable comment. As suggested by the reviewers, we conducted additional experiments to compare the particle size and aqueous solubility of BiOCl and Bi-DTPA. The results showed that Bi-DTPA readily formed a highly transparent solution in water, confirming its excellent solubility (**Fig. S1a**). In stark contrast, BiOCl quickly formed flocculent precipitates upon dispersion, indicating significantly lower water solubility than Bi-DTPA (**Fig. S1b**). Additionally, Bi-DTPA, being a small molecular compound with high solubility, is difficult to visualize under TEM (**Fig. S1c**). In contrast, TEM images revealed that BiOCl particles were composed of aggregated nanoparticles, a morphology consistent with their poor solubility (**Fig. S1d**). Consequently, BiOCl exhibited a much slower release rate from the BioGlue matrix, with the CT signal of BiOCl-BioGlue remaining stable for over 24 h. By comparison, the CT value of Bi-DTPA-BioGlue decreased markedly within 5 h, due to the rapid diffusion of Bi-DTPA from the hydrogel.

Revised text

...BiOCl, **composed of aggregated nanoparticles with poor water solubility**, was fabricated using a simple hydrolysis method (**Fig. S1d, e**), and Bi₂O₃ and Bi₂S₃ were obtained through commercial sources. (**Page 5**)

Revised figure

Fig. S1. Photographs of Bi-DTPA (a) and BiOCl (b) dispersions in water under static conditions. TEM image of Bi-DTPA (c) and BiOCl (d). (e) XRD pattern of BiOCl.

Comment 2:

The authors conclude that the CT imaging detection limit for Bi-BioGlue emboli is 1.2 mm. However, this conclusion raises several concerns: (1) In Figure 4a, the 1 mm embolus in red meat is still visible and distinguishable from the surrounding background, calling the 1.2 mm detection limit into question. (2) The authors should verify whether factors such as the amount of Bi doped in BioGlue, the depth of the Bi-BioGlue within the tissue, and the CT machine parameters could influence the detection limit. (3) Additionally, the authors should consider employing a more standardized and scientific method to confirm the detection limit, rather than relying solely on visual judgment.

Reply and corresponding changes:

Thank you very much for your valuable comments! As the reviewer suggested, we have revised and provided a detailed and specific description of the emboli detection limit, and systematically investigated the impacts of Bi doping concentration, embolus embedding depth, and key CT machine parameters on the embolus detection capability in the scanning results.

We systematically optimized the Bi doping concentration to assess its effect on the CT imaging performance of Bi-BioGlue. Increasing the Bi content enhanced the sensitivity of CT detection; however, excessive doping led to undesirable changes in the colloidal properties and raised potential concerns about biosafety. Our results showed that the CT value of bulk Bi-BioGlue gradually increased with rising Bi concentrations. When the Bi concentration reached 105 mM, the CT value of Bi-BioGlue reached approximately 659 HU, yielding a favorable signal-to-background ratio compared to normal tissues, which typically exhibit CT values in the range of 50-80 HU. It should be noted that maintaining a high CT value

in bulk Bi-BioGlue relative to soft tissue is essential, as the formation of very small BioGlue emboli will lead to a certain degree of CT signal attenuation due to partial volume effects. Importantly, at this concentration, the mechanical and adhesive properties of BioGlue were not adversely affected. Therefore, 105 mM was selected as the optimal Bi doping concentration for the following studies.

We also investigated the detection limit of Bi-BioGlue for emboli of different sizes embedded within red meat of varying thicknesses. *Ex vivo* CT imaging showed that emboli with diameters of 1.0 mm and 1.2 mm exhibited CT values of approximately 300 HU and 500 HU, respectively. These values are slightly lower than those of bulk samples due to partial volume effects. Notably, the CT values of Bi-BioGlue emboli were not significantly affected by the presence of varying thicknesses (1.0, 3.0, 5.0 cm) of overlying red meat, which is attributed to the excellent tissue penetration capability of CT imaging (**Fig. S12a**). *In vivo*, these values decreased to approximately 220 HU and 400 HU, respectively (**Fig. S13**). Due to its small size and a CT value only marginally higher than surrounding tissue, the 1.0 mm embolus posed significant challenges for reliable detection *in vivo*. In contrast, the 1.2 mm embolus exhibited a significantly stronger contrast against background tissues and could be clearly detected. Therefore, 1.2 mm was determined to be the minimum detectable embolus size *in vivo* using Bi-BioGlue.

Furthermore, we explored the influence of CT scanning parameters on embolus detectability, with particular attention to tube voltage and slice thickness. To achieve optimal visualization of ultra-small emboli, the slice thickness was set to the minimum value supported by the instrument (0.5 mm). When the tube voltage was varied between 80 and 140 kV, only minor fluctuations in CT values were observed across different embolus sizes, which can be attributed to the high K-edge energy of Bi (Fig. S12b). These results indicate that changes in scanning parameters do not significantly affect the detection limit, and that 1.2 mm remains the smallest embolus size that can be reliably detected under a range of CT imaging conditions.

Revised text and figures

Bi doping concentration was systematically optimized to assess its effect on the CT imaging performance of Bi-BioGlue. Increasing the Bi content enhanced the sensitivity of CT detection; however, excessive doping led to undesirable changes in the colloidal properties and raised potential concerns about biosafety. The result showed that the CT value of bulk Bi-BioGlue gradually increased with rising Bi concentrations. When the Bi concentration reached 105 mM, the CT value of Bi-BioGlue reached approximately 659 HU, yielding a favorable signal-to-background ratio compared to normal tissues, which typically exhibit CT values in the range of 50-80 HU. It should be noted that maintaining a high CT value in bulk Bi-BioGlue relative to soft tissue is essential, as the formation of very small BioGlue emboli will lead to a certain degree of CT signal attenuation due to partial volume effects. Therefore, 105 mM was selected as the optimal Bi doping concentration for the following studies...(Page 5)

The crosslinking kinetics of both BioGlue and Bi-BioGlue by rheological

analysis demonstrated that upon mixing, the BSA and glutaraldehyde components exhibited similarly rapid crosslinking kinetics, forming an opaque white solid within seconds (**Fig. 2h**) ... (Page 6)

Fig. 2. (h) Dynamic oscillatory time sweep measurement of BioGlue and Bi-BioGlue.

...The detection limit of Bi-BioGlue was investigated for emboli of different sizes embedded within red meat of varying thicknesses (**Fig. S12a**). *Ex vivo* CT imaging showed that emboli with diameters of 1.0 mm and 1.2 mm exhibited CT values of approximately 300 HU and 500 HU, respectively. These values are slightly lower than those of bulk samples due to partial volume effects. Notably, the CT values of Bi-BioGlue emboli were not significantly affected by the presence of varying thicknesses (1.0, 3.0, 5.0 cm) of overlying red meat, demonstrating the excellent tissue penetration capability of CT imaging.

Furthermore, we explored the influence of CT scanning parameters on embolus detectability, with particular attention to tube voltage and slice thickness. To achieve optimal visualization of ultra-small emboli, the slice thickness was set to the minimum value supported by the instrument (0.5 mm). When the tube voltage was varied between 80 and 140 kV, only minor fluctuations in CT values were observed across different embolus sizes, which can be attributed to the high K-edge energy of Bi (**Fig. S12b**, corresponding to **Fig. 4a**). These results indicate that changes in scanning parameters do not significantly affect the embolus detection capability. (Page 8)

Fig. S12. (a) CT images of Bi-BioGlue emboli in red meat with different thicknesses (0, 1.0, 3.0, 5.0 cm). (b) CT images of Bi-BioGlue emboli of different sizes in red meat under different voltages (80, 100, 120, 140 kV) and scanning layer thicknesses (0.5, 1.0, 2.0, 3.0, 4.0, 5.0, 10.0 mm).

Compared to *ex vivo* conditions, the CT values of 1.0 mm and 1.2 mm Bi-BioGlue emboli decreased to approximately 220 HU and 400 HU, respectively, *in vivo* (Fig. S13). Owing to its small size and a CT value only slightly higher than that of surrounding tissues, the 1.0 mm embolus was difficult to reliably identify *in vivo*. In contrast, the 1.2 mm embolus exhibited markedly enhanced contrast against background tissue and was clearly visualized. Therefore, 1.2 mm was determined to be the minimum detectable embolus size *in vivo* using Bi-BioGlue. (Page 10)

Fig. S13. CT images of 1.0 mm Bi-BioGlue emboli *in vivo*.

Ex Vivo CT imaging of Bi-BioGlue Emboli with Different Parameters. Bi-BioGlue was carefully trimmed into cubic emboli with diameters of 1.0, 1.2, 2.0, 3.0, 4.0, and 5.0 mm. CT imaging was conducted under various conditions to evaluate the detectability of emboli in *ex vivo* settings.

Scanning parameters optimization: emboli were imaged while embedded in 2 cm-thick red meat. Slice thicknesses were varied at 0.5, 1.0, 2.0, 3.0, 4.0, 5.0, and 10 mm, and the tube voltage was adjusted between 80, 100, 120, and 140 kV.

Impact of penetration depth on the detectability of Bi-BioGlue emboli: To assess the penetration capability of CT, 1.2 mm emboli were sandwiched between red meat slices of 1.0, 3.0, and 5.0 cm thickness. CT scanning was conducted with a FOV of 256×256 mm, slice thickness of 0.5 mm, and tube voltage of 120 kV.

Detection limits in air and tissue: Emboli were placed either in air or sandwiched between two 2 cm-thick slices of red meat. Scanning was performed with a FOV of 256×256 mm, a slice thickness of 0.5 mm, and adaptive tube current at 120 kV. (Pages 15-16)

Comment 3:

The authors used spectral CT to differentiate Bi-BioGlue from Ca-BioGlue. However, two additional control groups should be included for comparison: (1) BioGlue and Ca-BioGlue, and (2) Iodine-BioGlue and Ca-BioGlue. These controls would better demonstrate that only the developed Bi-BioGlue can effectively distinguish emboli from calcified tissues.

Reply and corresponding changes:

Thank you very much! As the reviewers suggested, we have supplemented the scanning images of 96-well plate spectral CT at different keV for comparisons among Ca-BioGlue, Iodine-BioGlue, and BioGlue, as depicted in the figure. 200 μ L of I-BioGlue, Bi-BioGlue, and BioGlue were placed in positions E3, E6, and E9 of the 96-well plate respectively, and Ca-BioGlue was added to the remaining holes. CT imaging results showed that Bi-BioGlue consistently maintained a high brightness across the entire range of X-ray energies, whereas BioGlue alone remained undetectable at all energy levels. Both I-BioGlue and Ca-BioGlue exhibited strong brightness at low X-ray energies, but their signals declined sharply as the X-ray energy increased, becoming extremely weak at higher energies. These findings clearly demonstrate that Bi-BioGlue enables effective differentiation between emboli and calcified tissues in spectral CT imaging (**Fig. S5a-d**).

Revised text

I-BioGlue, Bi-BioGlue, BioGlue injected into a well at row E and column 3, 6, 9 of 96-well plate, respectively, ... CT imaging results showed that Bi-BioGlue consistently maintained a high brightness across the entire range of X-ray energies, whereas BioGlue alone remained undetectable at all energy levels. Both I-BioGlue and Ca-BioGlue exhibited strong brightness at low X-ray energies, but their signals declined sharply as the X-ray energy increased, becoming extremely weak at higher energies (**Fig. 5c, d**). These findings clearly demonstrate that Bi-BioGlue enables effective differentiation between emboli and calcified tissues in spectral CT imaging. (Page 12)

Revised figure

Fig. 5. Spectral CT imaging for differentiation of Bi-BioGlue-induced embolus from confusable calcified nodule. **(a)** Schematic illustration of I-BioGlue injected into a well at row E and column 3, Bi-BioGlue injected into a well at row E and column 6, BioGlue injected into a well at row E and column 9 of 96-well plate, while filling other wells with $\text{Ca}_3(\text{PO}_4)_2$ -doped BioGlue. Conventional CT images **(b)** and spectral CT images **(c)** of 96-well plate. **(d)** Spectral CT signal curves of Bi-BioGlue, I-BioGlue, BioGlue and $\text{Ca}_3(\text{PO}_4)_2$ -doped BioGlue at different monochromatic energies. ... (Page 11)

Comment 4:

In Figures 5e and 5f, the authors should clarify what the two types of arrows represent.

Reply and corresponding changes:

Thank you very much! We sincerely apologize for omitting the two types of arrows represent in Figures 5e and 5f. A detailed description about the two types of arrows have been provided in revised manuscript, which the thin arrows herein indicate Ca-BioGlue simulated pulmonary calcification, and the wide arrows signify a pulmonary embolism caused by Bi-BioGlue.

Revised text

Fig. 5. ...The calcified nodule is indicated by a thin arrow, and the Bi-BioGlue embolus is indicated by a wide arrow... (Page 11)

Comment 5:

The flow of results is disjointed. The authors first discuss Figure 4a, then move to Figure 5a-d, before returning to Figure 4b-g, and finally return to Figure 5e-h. This organization is confusing and should be reconsidered for better coherence.

Reply and corresponding changes:

Thank you very much for your reminder! We have revised the manuscript to reorganize the figure sequence and corresponding text for improved logical flow and clarity.

Comment 6:

The authors should provide more information on how long it takes for BioGlue to form a pulmonary embolism *in vivo*.

Reply and corresponding changes:

Thanks for your valuable suggestions! In clinical practice, once BioGlue detaches at the vascular adhesion site, the detached emboli would enter the bloodstream, quickly obstruct branching blood vessels and cause insufficient blood supply to organs. To simulate the clinical embolus detachment and vascular embolism, we established a rat pulmonary embolism model by injecting emboli *via* the posterior vena cava puncture. The emboli were gelled *in vitro* and punctured into the posterior vena cava of rats. Once entering the bloodstream, the emboli would flow with blood into pulmonary circulation and rapidly occlude the pulmonary artery within seconds or minutes. This information has been added to the revised manuscript.

Revised text

...Once entering the bloodstream, the emboli would flow with blood into pulmonary circulation and rapidly occlude the pulmonary artery within seconds or minutes... (Page 9)

Comment 7:

The rationale for using H9c2 cells, a cardiomyocyte cell line, for the biosafety evaluation should be explained, as this cell line may not be the most appropriate model for assessing BioGlue's general biocompatibility.

Reply and corresponding changes:

Thanks for your valuable suggestions! In clinic practice, BioGlue is prevalently employed in various cardiac procedures, such as proximal aortic dissection repair, aortic root and arch reconstruction, valve repair and replacement surgery. H9c2 cells are one of the most used cell lines in cardiovascular system research, which were chosen for biosafety evaluation in this study. Meanwhile, as the reviewers suggest, another commonly used cell line in cardiovascular research, human umbilical vein endothelia line (HUVEC cells), was employed in this study to evaluate the safety of the Bi- BioGlue. The cell viabilities of H9c2 cells and HUVEC cells were remained higher than 80% even at a concentration as high as 20 mg/mL Bi-BioGlue for 48 h, which results suggested low cytotoxicity of Bi-BioGlue towards both kinds of cells (Fig. 6b).

Revised text

...The *in vitro* cytotoxicity was investigated by standard MTT assay. H9c2 cells and HUVEC cells, two of the most commonly used cell lines in cardiovascular research, were cultured in media containing various concentrations of Bi-BioGlue for 24 and 48 h. The cell viabilities of H9c2 cell and HUVEC cells lines remained higher than 80% even at a concentration as high as 20 mg/mL Bi-BioGlue which results

suggested low cytotoxicity of Bi-BioGlue towards H9c2 cells and HUVEC cells (Fig. S16a, S16b). (Page 12)

Revised figure

Fig. S16. Biosafety evaluation of Bi-BioGlue. Cell viabilities of H9c2 cells (a) and HUVEC cells (b) after incubated with different concentrations of Bi-BioGlue for 24 or 48 h. (c) Body weight changes of control group and Bi-BioGlue group in 45 days. Bio-chemical analysis (d) and routine blood tests (e) of control group, and Bi-BioGlue group in 45 days. (f) H&E staining images of major organs (heart, liver,

spleen, lung, and kidney) collected from the rats in control group and Bi-BioGlue group in 45 days.

Response to Reviewer 2

Comments:

The manuscript describes the characterization of a bismuth labelled derivative of a clinically used surgical adhesive, and preclinical in vivo assessment of the feasibility to visualize vascular emboli using this compound using CT imaging. The authors show that the bismuth labelled derivative has the same adhesive capabilities compared to the FDA approved BioGlue.

Comment 1:

In the introduction the size of emboli when damage occurs needs to be mentioned and related to the size detected in the results. The authors also need to explain the benefit of hypersensitive detection of millimetre sized emboli. How do the authors envision clinical application? Through watch and wait or intervention therapy? The extent of symptoms will be dependent on the size of the embolism. Will there be differences in treatment compared to standard embolisms? As anticoagulants will not work and nor will fibrinolytic therapy.

Reply and corresponding changes:

Thank you very much! The clinical impact of emboli is closely related to their size. Large emboli (>5.0 mm) can cause life-threatening events such as pulmonary embolism or major stroke. Medium-sized emboli (1.0-5.0 mm) may lodge in smaller vessels of the lungs, brain, kidneys or retina, leading to localized ischemia and organ dysfunction. Even micro emboli around 1 mm, though initially asymptomatic, may trigger chronic inflammation and cumulative damage, especially in sensitive organs. Since emboli of various sizes can pose significant safety risks, it is essential to detect emboli across a wide size range. Therefore, there is an urgent need for highly sensitive detection methods, particularly those capable of identifying micro emboli as small as 1 mm in diameter.

Timely detection and early intervention of emboli significantly improve patient outcomes. During the use of surgical adhesives such as BioGlue, the intraoperative period represents the most critical window for embolus formation due to potential detachment of adhesive fragments. Therefore, immediate post-operative imaging to detect dislodged emboli is essential. Moreover, the early postoperative period also carries a high risk for delayed embolus formation, highlighting the need for continued surveillance. If translated into clinical practice, our proposed strategy would follow a clear detection timeline: a whole-body CT scan should be performed immediately after surgery to screen for intraoperatively dislodged emboli, followed by regular CT scans during the early postoperative period to identify any subsequently formed emboli. This time-defined imaging protocol ensures that even ultra-small emboli can

be detected at high-risk stages, supporting timely clinical intervention.

Anticoagulant and fibrinolytic therapy are important approaches for treating thrombosis, as they work by affecting platelets and related coagulation factors. In clinical practice, a range of therapeutic strategies are available for different types of emboli, including pharmacological thrombolysis, anticoagulant therapy, interventional thrombectomy, and surgical embolectomy. Thrombolytic therapy is commonly used for acute thrombotic events (such as acute ischemic stroke or pulmonary embolism), functioning by activating the body's fibrinolytic system to dissolve the thrombus. Anticoagulant therapy is widely employed in the prevention and management of thromboembolic disorders such as venous thromboembolism and atrial fibrillation-related thrombi, primarily by inhibiting the coagulation cascade and preventing thrombus propagation. Interventional thrombectomy is typically applied in cases of acute large-vessel occlusion or when pharmacological therapies prove insufficient, enabling rapid restoration of blood flow. Surgical embolectomy is generally reserved for massive or life-threatening emboli, such as severe pulmonary or peripheral arterial embolism.

However, BioGlue is primarily composed of glutaraldehyde and cross-linked proteins, resulting in emboli with highly stable and degradation-resistant structures that are unresponsive to conventional anticoagulant or thrombolytic therapies. At present, interventional thrombectomy or surgical embolectomy are the only effective clinical options for removing BioGlue-derived emboli. Failure to detect and treat such emboli in a timely manner may lead to serious health consequences or life-threatening complications. Therefore, early detection and prompt intervention for BioGlue embolism are of critical importance in clinical management. This information has been added to the revised manuscript.

Revised text

Adhesive-derived emboli with different sizes (ranging from 1.0 mm to several millimeters) ... Besides, adhesive-derived emboli cannot be effectively treated by conventional anticoagulant or thrombolytic therapies and typically require interventional or surgical removal. (Page 2)

Comment 2:

A- Line 48: Optical imaging is not a suitable imaging methodology for this application due to its limited penetration depth and should be excluded here.

Reply and corresponding changes:

Thank you very much for your suggestions! We have removed the description of optical imaging from the introduction in the revised manuscript.

Comment 3:

- Line 53: The authors present the bismuth labelling as novel, but this has already been applied for adhesives. Tian et al (Advanced materials 2023) describe a bismuth-pectin organic-inorganic complex that was engineered into a transformable

microgel (that they call bi-glue) and that is compatible with x-ray imaging. The statement made by the authors needs to be revised and the paper by Tian et al should be accurately referenced and the differences in approaches should be discussed.

Reply and corresponding changes:

Thank you very much! Tian et al. reported a bismuth-pectin organic-inorganic complex as an innovative contrast agent for gastrointestinal X-ray and CT imaging. Their interesting work addresses a key limitation of conventional X-ray CT contrast agents, which lack adhesion to gastrointestinal tissues and are rapidly cleared after oral administration, preventing complete imaging of the gastrointestinal tract. The bismuth-pectin complex developed by Tian et al. exhibits certain gastrointestinal adhesive properties, facilitating comprehensive X-ray and CT imaging throughout the digestive tract.

Although the material is named "bi-glue", the transformable microgel reported in the referenced study is fundamentally a hydrogel system, not a surgical adhesive in the conventional or clinical sense. The adhesion of the bismuth-pectin complex merely slows the transit of the material through the gastrointestinal tract without physically bonding different tissues together. Moreover, if such contrast agents were to possess the ability to bond disparate tissues like BioGlue, they would likely cause severe adverse effects such as gastrointestinal adhesions, which are clinically unacceptable.

In contrast, the Bi-BioGlue we propose is a true surgical adhesive capable of firmly bonding different tissues and is specifically designed for detecting emboli caused by BioGlue detachment. Therefore, these two materials differ fundamentally in their research focus, scientific problems addressed, material properties, and application scenarios. We have accurately cited this study and used it as a representative example to describe its application area.

Revised text

Incorporating imaging units into adhesives or gels is a commonly employed strategy for visualizing them through various imaging techniques like computed tomography (CT) imaging¹⁶⁻¹⁹, magnetic resonance imaging²⁰, nuclear imaging^{17, 21}, ultra-sound imaging²²⁻²⁴, and photoacoustic imaging²⁵⁻²⁷. This approach has been extensively utilized to detect the location, morphology, and degradation behavior of adhesives and gels, and has found widespread application in tissue engineering, **disease diagnosis**, drug delivery, and immunotherapy. (Page 3)

Comment 4:

Fig 3 shows the visualization of emboli over the course of 42 days. What was the percentage of resorbance of emboli? If all were resorbed, what is then the additional value of early detection?

Reply and corresponding changes:

Thank you very much! BioGlue is one of the most widely used surgical

adhesives and is gradually resorbed *in vivo* over a period of months to even years. Therefore, Bi-BioGlue fixed at the wound site is inherently safe and will eventually be metabolized by the body (*J. Thorac. Cardiovasc. Surg.*, **2019**, 157, 176). Figure 3 shows Bi-BioGlue fixed at the wound site after surgical adhesion, rather than representing detached emboli. Our 42-day monitoring study also demonstrated its gradual degradation and resorption over time. However, the situation is entirely different for emboli. Once an embolus forms, it can immediately cause tissue damage or even life-threatening complications. As such, relying on the eventual resorption of adhesive-derived emboli is not a viable strategy. Instead, it is imperative to detect these emboli at an early stage and promptly remove them through interventional or surgical procedures.

Comment 5:

An explanation should be provided to how detection of millimeter sized emboli will significantly improve the biosafety of surgical adhesions. In my opinion these two features are not directly linked as the biosafety profile of the adhesive is dependent on the chemical composition itself and not on its detectability. Figure 6 shows biosafety assessment of Bi-BioGlue without comparison to the original BioGlue, therefore significant improvement cannot be claimed.

Reply and corresponding changes:

The biosafety of a surgical adhesive is not solely determined by its chemical composition; its potential to form undetectable emboli also poses a significant safety risk. Therefore, the overall biosafety of an adhesive depends both on the biocompatibility of its constituents and on the ability to detect and intervene in embolic events in a timely manner.

BioGlue is primarily composed of bovine serum albumin and glutaraldehyde, and the albumin is inherently biocompatible. Although glutaraldehyde has known chemical toxicity, it forms covalent bonds with proteins and tissues during application, becoming immobilized within the adhesive matrix, and is then slowly metabolized over time. As a result, its intrinsic chemical toxicity exerts minimal biological impact, which contributes to the high chemical safety profile of BioGlue and supports its FDA approval and widespread clinical use.

However, the main biosafety concern lies in the potential formation of emboli caused by dislodged BioGlue fragments, which are difficult to detect with conventional methods. This embolic risk represents a recognized clinical challenge in the use of BioGlue. Therefore, in this study, we address this issue by incorporating BiOCl into BioGlue, enabling the real-time detection of even sub-millimeter emboli via CT imaging. This strategy significantly reduces the potential risks associated with embolism and enhances the overall safety profile of BioGlue.

In terms of chemical composition, our development of Bi-BioGlue does not alter the intrinsic biosafety profile of the original BioGlue. However, from the perspective of mitigating the risk of adverse events caused by dislodged emboli, Bi-BioGlue offers a significant safety advantage. While conventional BioGlue lacks the ability to

visualize emboli once detached, Bi-BioGlue enables timely detection of emboli through CT imaging, allowing for prompt clinical intervention. Therefore, our approach improves the overall safety of BioGlue not by changing its chemical composition but by solving one of its most critical clinical challenges, which is the inability to detect embolic complications. By enabling reliable post-operative monitoring of emboli, Bi-BioGlue provides a more comprehensive and proactive safety profile. Thus, our approach provides a scientifically grounded and rational solution to improve the biosafety of BioGlue through the integration of embolus-traceable contrast functionality.

Regarding Figure 6, our primary objective was to evaluate whether the incorporation of BiOCl introduces any conventional biosafety concerns, including effects on body weight, serum biochemistry, hematology, and histopathology. The comprehensive toxicological evaluation provided strong evidence supporting the excellent biocompatibility of BiOCl. However, we would like to emphasize that conventional biosafety assessment is not the main focus of our study. This work aims to solve the critical clinical challenge of detecting adhesive emboli caused by BioGlue detachment, a risk that is currently difficult to manage. The significant improvement in biosafety we claim is not based on standard toxicity profiles, but rather on the ability of Bi-BioGlue to enable highly sensitive and timely detection of emboli, which directly contributes to reducing embolism-related risks.

This information has been added to the revised manuscript.

Revised text

...Therefore, our approach enhances BioGlue's safety by addressing the clinical challenge of undetectable embolic complications, rather than altering its chemical composition. By enabling reliable post-operative monitoring of emboli, Bi-BioGlue provides a more comprehensive and proactive safety profile. (Page 10)

...The systematic toxicological assessment demonstrated Bi-BioGlue does not alter the intrinsic biosafety profile of the original BioGlue in terms of chemical composition. (Page 13)

Comment 6:

Stability assessment of the created complex is insufficient. How can the authors ensure that the complex will remain stable *in vivo*? This should at least be tested under comparable conditions.

Reply and corresponding changes:

Thank you very much! We systematically studied the morphological changes of Bi-BioGlue after incubation in various simulated physiological media over different time periods, and assessed the release of Bi ions. The experimental results demonstrated that Bi-BioGlue maintained a stable morphology throughout the monitoring period with negligible Bi³⁺ release. These findings indicate that BiOCl exhibits efficient labeling within the BioGlue matrix, and the resulting Bi-BioGlue possesses excellent physicochemical stability.

In vivo experiments further revealed a gradual decline in the CT signal of Bi-BioGlue over time, indicating a slow degradation process in physiological environments. This slow degradation is primarily attributed to the continuous movement of body fluids, which promotes material transport and diffusion-consistent with the intrinsic long-term degradation behavior of BioGlue itself. Notably, this gradual degradation provides a sufficient imaging window for the detection of emboli both intraoperatively and during the early postoperative high-risk period.

In contrast, other modified adhesives such as Bi-DTPA-BioGlue and I-BioGlue exhibit rapid release of contrast agents, resulting in a sharp decline in CT signal intensity, thereby failing to support stable embolus imaging and timely detection. In summary, we have provided a more accurate description of the stability of Bi-BioGlue: it exhibits high labeling efficiency and adequate *in vivo* stability, enabling a reliable imaging time window for effective detection of emboli.

Revised text

...In addition, the morphological changes of Bi-BioGlue after incubation in various simulated physiological media over different time periods and the release of Bi ions were systematically studied. The experimental results demonstrated that Bi-BioGlue maintained a stable morphology throughout the monitoring period, with negligible Bi^{3+} release (Fig. S5, S6). ...These findings indicate that BiOCl exhibits efficient labeling within the BioGlue matrix, and the resulting Bi-BioGlue possesses excellent physicochemical stability. (Page 6)

Revised figure

Fig. S6. The release curves of Bi elements in Bi-BioGlue within 14 days across different media including H₂O, PBS, H₂O₂, and serum.

Comment 7:

How is the apparent degradation of the adhesive being taken into account? It needs to be investigated that the Bi remains complexed during degradation. The

presence of non-labelled adhesive and a possible effect on emboli formation and visualization needs to be assessed.

Reply and corresponding changes:

Thank you very much for your constructive comments. BiOCl was selected as a high-performance imaging label due to its favorable physicochemical properties, enabling uniform labeling of BioGlue. *In vitro* studies, including CT imaging and elemental mapping, confirmed the homogeneous distribution of BiOCl within the BioGlue matrix, with no evidence of unlabeled regions. Stability assessments further demonstrated that BiOCl was not rapidly released from Bi-BioGlue, indicating strong retention of the BiOCl component. During *in vivo* application, the two primary components, BiOCl-BSA mixture and glutaraldehyde, rapidly reacted upon contact, forming a solidified Bi-BioGlue within seconds. Although BioGlue inherently undergoes slow degradation *in vivo*, *in vivo* CT imaging revealed that Bi-BioGlue maintained stable morphology and size within the first week, with only a gradual decrease in CT signal intensity. This indicates that the initial degradation of Bi-BioGlue is very slow and that there is no apparent presence of unlabeled adhesive. As time progressed, Bi-BioGlue exhibited further degradation, as evidenced by a gradual reduction in size on CT imaging. Given the negligible Bi³⁺ release of BiOCl under simulated physiological conditions *in vitro*, it is reasonable to infer that BiOCl retains considerable labeling efficiency even during later stages of degradation.

Considering that the intraoperative and early postoperative periods represent high-risk windows for embolus detachment when using BioGlue, the demonstrated *in vivo* labeling efficiency of Bi-BioGlue during the early stage ensures the reliable detection of emboli within this critical timeframe.

Revised text

...The Bi-BioGlue maintained stable morphology and size within the first week, with only a gradual decrease in CT signal intensity. This indicates that the initial degradation of Bi-BioGlue is very slow and that there is no apparent presence of unlabeled adhesive. As time progressed, Bi-BioGlue exhibited further degradation, as evidenced by a gradual reduction in size on CT imaging, eventually the CT value of adhesion site...Given the negligible Bi³⁺ release of BiOCl under simulated physiological conditions *in vitro*, it is reasonable to infer that BiOCl retains considerable labeling efficiency even during later stages of degradation (**Fig. S11**). Considering that the intraoperative and early postoperative periods represent high-risk windows for embolus detachment when using BioGlue, the demonstrated *in vivo* labeling efficiency of Bi-BioGlue during the early stage ensures the reliable detection of emboli within this critical timeframe. (Page 8)

Revised figure

Fig. S11. The release curves of Bi elements in BiOCl within 14 days across different dispersion systems including H₂O, PBS, H₂O₂, and serum.

Response to Reviewer 3

The authors present a bismuth-based BiOCl-doped BioGlue for detecting vascular emboli originating from surgical adhesives using CT imaging. This CT-visible adhesive enables in vivo tracking of adhesive degradation and the real-time detection of small emboli. The Bi-BioGlue demonstrates enhanced stability, biocompatibility, and prolonged imaging persistence, with the capability to detect emboli as small as 1.2 mm in a pulmonary embolism model. Overall, this study is well-designed with a rigorous methodology. However, several questions and suggestions need to be addressed before considering publication.

Reply:

Thank you very much for your such positive comments!

Comment 1:

The authors used known molecules to form a crosslinked glue. However, the chemistry should be shown more clearly, including the chemical structures of each molecule and the detailed characterizations after synthesis.

Reply and corresponding changes:

Thank you very much! Based on the reviewer's suggestions, we have presented the chemical structure of each molecule in the figure. Additionally, BioGlue is composed of purified bovine serum albumin (45%, w/v) and glutaraldehyde (10%, w/v), which are mixed in a dual-barrel syringe and applied to the surgical site. The principle behind BioGlue is that glutaraldehyde forms a stable seal by covalently linking to the lysine amino acid fragment in bovine serum albumin, and it can also

connect to lysine residues in other proteins in the tissue, resulting in excellent tissue adhesion (Figure 2a).

Revised text

...BioGlue is composed of purified bovine serum albumin (45%, w/v) and glutaraldehyde (10%, w/v), which are mixed in a dual-barrel syringe and applied to the surgical site. The principle behind BioGlue is that glutaraldehyde forms a stable seal by covalently linking to the lysine amino acid fragment in bovine serum albumin, and it can also connect to lysine residues in other proteins in the tissue, resulting in excellent tissue adhesion. (Page 5)

Revised figure

Fig. 2. (a) Synthetic scheme of the Bi-doping BioGlue through the GA-induced cross-linking method.

Comment 2:

The cross-linking mechanism between BiOCl and the BioGlue matrix is briefly mentioned, but more experimental details are required. This should include the kinetics and degree of cross-linking, as these factors could directly influence adhesion strength and stability. Additionally, as BiOCl stability is crucial for sustained imaging, please discuss the stability of BiOCl in the adhesive under physiological pH and oxidative conditions.

Reply and corresponding changes:

Thank you very much! Bi-BioGlue and BioGlue share the same crosslinking mechanism, which is based on the covalent interaction between BSA and GA. The only difference lies in the composition of the protein component: Bi-BioGlue incorporates BSA doped with BiOCl, whereas BioGlue utilizes native BSA. Notably, BiOCl is physically encapsulated within the BioGlue matrix and does not participate in the chemical crosslinking process.

As suggested by the reviewers, we systematically evaluated the crosslinking kinetics of both BioGlue and Bi-BioGlue by rheological analysis. The results demonstrated that upon mixing, the BSA and glutaraldehyde components exhibited

similarly rapid crosslinking kinetics, forming an opaque white solid within seconds. Furthermore, tissue adhesion tests showed that Bi-BioGlue exhibited comparable adhesive strength to BioGlue. These findings indicate that the incorporation of BiOCl does not interfere with the crosslinking behavior or adhesive performance of BioGlue.

As suggested by the reviewer, we evaluated the structural stability of Bi-BioGlue under different simulated physiological conditions. Specifically, both Bi-BioGlue and BiOCl were incubated in four representative media: deionized water (H₂O), physiological buffer (10 mM PBS, pH 7.4), an oxidative stress environment (1 mM H₂O₂), and serum. At predetermined time points, we performed X-ray diffraction (XRD) analysis and quantified the release of Bi³⁺ ions to assess structural integrity and potential degradation. The results showed that BiOCl, whether incorporated into Bi-BioGlue or not, retained its crystalline structure across all media without noticeable changes. Moreover, the release of Bi³⁺ ions remained below 2% in all conditions. Additionally, the morphology and dimensions of Bi-BioGlue exhibited no obvious alterations during the incubation period. Collectively, these findings demonstrate the excellent structural and chemical stability of both BiOCl and Bi-BioGlue under physiologically relevant environments.

Revised text and figures

...Bi-based BioGlue and BioGlue share the same crosslinking mechanism, which is based on the covalent interaction between BSA and GA. The only difference lies in the composition of the protein component: Bi-based BioGlue incorporates BSA doped with Bi-based imaging probes, whereas BioGlue utilizes native BSA... (Page 5)

The crosslinking kinetics of both BioGlue and Bi-BioGlue by rheological analysis demonstrated that upon mixing, the BSA and glutaraldehyde components exhibited similarly rapid crosslinking kinetics, forming an opaque white solid within seconds (**Fig. 2h**). Further XRD analysis showed that BiOCl, whether incorporated into Bi-BioGlue or not, retained its crystalline structure across all media without noticeable changes (**Fig. S7**). (Page 6)

Fig. S7. XRD patterns of Bi-BioGlue after incubation in different media: H₂O (a), PBS (b), H₂O₂ (c), and serum (d) on days 1, 7, and 14. XRD patterns of BiOCl after incubation in different dispersion systems: H₂O (e), PBS (f), H₂O₂ (g), and serum (h)

on days 1, 7, and 14.

Comment 3:

On page 6, line 105, please include a reference for the synthesis method of Bi-DTPA.

Reply and corresponding changes:

Thank you very much! The reference for the synthesis of Bi-DTPA has been cited in the revised manuscript.

Comment 4:

In Fig S2, please add or label BioGlue, gelatin, and sodium alginate within the figures.

Reply and corresponding changes:

Thank you very much for your suggestion! BioGlue, gelatin, and alginate have been marked in the figures of the revised supporting information.

Revised figure

Fig. S2. Photographs of BiOCl-doped BioGlue, Chitosan-gelatin and sodium alginate-Ca²⁺ with different concentrations (0-88 mM) of BiOCl.

Comment 5:

In Fig. 2g, please clarify what the colors represent.

Reply and corresponding changes:

Thank you very much for your advice! We clarified the element corresponding to each color. In Fig. 2c, orange represents Cl element, yellow represents Bi element, and purple represents S element. The element mapping image clearly shows the

distribution of Cl, Bi, and S elements in Bi-BioGlue.

Revised text

Elemental mapping images demonstrated the presence and uniform distribution of Cl (orange), Bi (yellow), and S (purple) in the freeze-dried Bi-BioGlue (Fig. 2g). (Page 6)

Comment 6:

In Fig. 2k, what are the concentrations of Bi-BioGlue and iohexol-doped BioGlue used in this study? If the concentration is 105 mM, please explain the rationale for this choice. Additionally, please provide a more detailed comparison of detection sensitivity with standard iodine-based agents.

Reply and corresponding changes:

Thank you very much! We systematically optimized the Bi doping concentration to assess its effect on the CT imaging performance of Bi-BioGlue. Increasing the Bi content enhanced the sensitivity of CT detection; however, excessive doping led to undesirable changes in the colloidal properties and raised potential concerns about biosafety. Our results showed that the CT value of bulk Bi-BioGlue gradually increased with rising Bi concentrations. When the Bi concentration reached 105 mM, the CT value of Bi-BioGlue reached approximately 659 HU, yielding a favorable signal-to-background ratio compared to normal tissues, which typically exhibit CT values in the range of 40-80 HU. It should be noted that maintaining a high CT value in bulk Bi-BioGlue relative to soft tissue is essential, as the formation of very small BioGlue emboli will lead to a certain degree of CT signal attenuation due to partial volume effects. Importantly, at this concentration, the mechanical and adhesive properties of BioGlue were not adversely affected. Therefore, 105 mM was selected as the optimal Bi doping concentration for the following studies.

Indeed, we had provided a detailed comparison of detection sensitivity with standard iodine-based agents in original manuscript. On conventional CT, though CT intensity of both Bi-BioGlue and I-BioGlue was linear enhancement with the increasement of concentration, the Bi-BioGlue yielded a higher CT intensity compared to I-BioGlue on same Bi/I-doped concentration at commonly used tube voltage in clinic (120 kV) (Fig. 2e and Fig. S8). During spectral CT scanning, Due to the much higher K edge energy value of Bi (91 keV) compared to the I (33.2 keV), the CT intensity of I-BioGlue showed sharply attenuation decay curve and at high monochromatic X-ray energy on spectral CT imaging, while the Bi-BioGlue exhibited a relatively flat decay curve. At 190 KeV, the CT values of Bi-BioGlue was 3.9 times higher than that of iohexol-doped BioGlue. In terms of doping stability, since clinical iodine-based contrast agents are small molecules with excellent water solubility, I-BioGlue exhibited a burst release behavior of imaging probes similar to that of Bi-DTPA-BioGlue when immersed in water. These results indicate that clinical iodine-based contrast agents are unsuitable for labeling surgical adhesives due to their relatively poor sensitivity and labeling capability.

Revised text

Bi doping concentration was systematically optimized to assess its effect on the CT imaging performance of Bi-BioGlue. Increasing the Bi content enhanced the sensitivity of CT detection; however, excessive doping led to undesirable changes in the colloidal properties and raised potential concerns about biosafety. The result showed that the CT value of bulk Bi-BioGlue gradually increased with rising Bi concentrations. When the Bi concentration reached 105 mM, the CT value of Bi-BioGlue reached approximately 659 HU, yielding a favorable signal-to-background ratio compared to normal tissues, which typically exhibit CT values in the range of 50-80 HU. It should be noted that maintaining a high CT value in bulk Bi-BioGlue relative to soft tissue is essential, as the formation of very small BioGlue emboli will lead to a certain degree of CT signal attenuation due to partial volume effects. Therefore, 105 mM was selected as the optimal Bi doping concentration for the following studies. (Page 5)

Comment 7:

In Fig. S3, the color change in DMEM from D0 to D14 (red to yellow) likely reflects pH changes. Please provide an explanation of any chemical reactions or degradation processes that may have occurred during the time period.

Reply and corresponding changes:

Thank you very much! In accordance with the reviewer's suggestions, we repeated the DMEM static stability experiment under more stringent and rigorous conditions, and included a BioGlue group for comparison. The color of DMEM incubated with Bi-BioGlue gradually lightened over 14 days, without showing any signs of yellowing. A similar color change was also observed in DMEM incubated with BioGlue. It is worth noting that in our previous experiments, yellowing of DMEM was observed when incubated with Bi-BioGlue over the same period, which was most likely caused by bacterial contamination. These findings indicate that the color change of DMEM is unrelated to the presence of BiOCl. In addition, we measured the pH of the solutions and found no significant changes. Given that phenol red, the pH indicator in DMEM, does not chemically react with any components of Bi-BioGlue, we infer that the lightening of DMEM color is primarily caused by the physical adsorption of phenol red by Bi-BioGlue.

Revised figure

Fig. S5. Photographs of Bi-BioGlue immersed in PBS, normal saline and DMEM for 14 days. Photographs BioGlue immersed in DMEM for 14 days.

Comment 8:

On page 7, line 141, verify if Fig. 1c is correctly referenced.

Reply and corresponding changes:

Thank you very much for pointing out the error! We apologize for the mistake made here and have corrected the erroneously cited Fig. 1c to Fig. 2e.

Comment 9:

In Fig. S6, please provide experimental details, including bonding duration and the time elapsed before testing Bi-BioGlue.

Reply and corresponding changes:

Thank you very much! To evaluate the tissue adhesive ability of Bi-BioGlue, we investigated its adhesive properties on pig aorta vessels *ex vivo*. Bi-BioGlue was injected into the leakage site using a syringe, with an adhesion time of 30 s. After the adhesion, the system was allowed to stabilize for 120 s before red ink, simulating blood, was injected from the upper end. No "blood" was observed to leak, thereby confirming the excellent adhesive capability of Bi-BioGlue.

Revised text

***Ex Vivo* Tissue Adhesion of Bi-BioGlue.** To evaluate the tissue adhesive performance of Bi-BioGlue, an *ex vivo* leakage model was established using isolated pig aorta vessels. A longitudinal incision was made on the vessel wall to simulate a leakage site. Bi-BioGlue was applied to the defect using a syringe, followed by 30 s of contact to initiate adhesion. The repaired vessel was then allowed to stand undisturbed for 120 s to ensure complete gelation and bonding. Subsequently, red ink was injected from the proximal end of the vessel to simulate blood flow. The absence of ink leakage from the repair site was used as an indicator of effective tissue adhesion. (Page 14)

Comment 10:

In Fig. 3f, indicate the time required to achieve sealing and hemostatic effects.

This information should be included in the experimental section or figure legend.

Reply and corresponding changes:

Thank you for your attention to this issue. Both BioGlue and Bi-BioGlue can seal and stop liver bleeding within about 30 s, and the time of hemostasis has been marked in the experimental section.

Revised text

...Both adhesives sealed the bleeding site and stopped hemorrhage within approximately 30 s... (Page 15)

Comment 11:

In Fig. 4a, provide more details regarding the preparation of Bi-BioGlue emboli in air and in red meat.

Reply and corresponding changes:

Thank you very much! We conducted an investigation into the limits of CT imaging detection of thrombi in air and red meat. First, the gel was allowed to polymerize in a petri dish, after which it was carefully trimmed into cubic gel blocks of varying sizes. Then, we placed the emboli either in air or sandwiched between two slices of red meat with defined thickness for CT scanning.

Revised text

Bi-BioGlue was carefully trimmed into cubic emboli with diameters of 1.0, 1.2, 2.0, 3.0, 4.0, and 5.0 mm. CT imaging was conducted under various conditions to evaluate the detectability of emboli in ex vivo settings. (Page 15)

Detection limits in air and tissue: Emboli were placed either in air or sandwiched between two 2 cm-thick slices of red meat. Scanning was performed with a FOV of 256×256 mm, a slice thickness of 0.5 mm, and adaptive tube current at 120 kV. (Page 16)

Comment 12:

In Fig. 5a, describe the process for creating $\text{Ca}_3(\text{PO}_4)_2$ -doped BioGlue.

Reply and corresponding changes:

Thank you very much! The $\text{Ca}_3(\text{PO}_4)_2$ -doped BioGlue is prepared as follows. First, 4.5 g of $\text{Ca}_3(\text{PO}_4)_2$ and 4 g of BSA are weighed and dissolved in water to form $\text{Ca}_3(\text{PO}_4)_2$ /BSA mixture (final volume 10 mL, $\text{Ca}_3(\text{PO}_4)_2$ 1.45 mol/L, BSA 400 mg/mL), followed by stirring evenly. It is worth noting that during the preparation of calcium phosphate-doped BioGlue, the concentration of BSA used was 400 mg/mL rather than 450 mg/mL. This adjustment was made because a higher BSA concentration, combined with calcium phosphate, is difficult to uniformly disperse in water. Since the CT contrast primarily arises from the calcium content, the slight reduction in BSA concentration is not expected to affect the experimental outcomes.

Next, $\text{Ca}_3(\text{PO}_4)_2/\text{BSA}$ -doped BioGlue was formed by reacting a mixture of $\text{Ca}_3(\text{PO}_4)_2/\text{BSA}$ and 10% glutaraldehyde in a 96-well plate, with volumes of 160 $\mu\text{L}/\text{well}$ and 40 $\mu\text{L}/\text{well}$, respectively.

Revised text

...4.5 g of $\text{Ca}_3(\text{PO}_4)_2$ and 4 g of BSA are weighed and dissolved in water to form $\text{Ca}_3(\text{PO}_4)_2/\text{BSA}$ mixture (final volume 10 mL, $\text{Ca}_3(\text{PO}_4)_2$ 1.45 mol/L, BSA 400 mg/mL), followed by stirring evenly. It is worth noting that during the preparation of $\text{Ca}_3(\text{PO}_4)_2$ -doped BioGlue, the concentration of BSA used was 400 mg/mL rather than 450 mg/mL. This adjustment was made because a higher BSA concentration, combined with calcium phosphate, is difficult to uniformly disperse in water. Since the CT contrast primarily arises from the calcium content, the slight reduction in BSA concentration is not expected to affect the experimental outcomes. Next, $\text{Ca}_3(\text{PO}_4)_2/\text{BSA}$ -doped BioGlue was formed by reacting a mixture of $\text{Ca}_3(\text{PO}_4)_2/\text{BSA}$ and 10% glutaraldehyde in a 96-well plate, with volumes of 160 $\mu\text{L}/\text{well}$ and 40 $\mu\text{L}/\text{well}$, respectively... (Page 16)

Comment 13:

On page 15, line 317, ensure consistency in units, as “1.2 mm” appears in several places. Please clarify if this measurement refers to diameter or volume.

Reply and corresponding changes:

Thank you very much! I apologize for the mistake in the text. We clarified that the term "1.2 mm" throughout the manuscript refers to the dimension (size) rather than the volume.

Revised text

...*Ex vivo* CT imaging showed that emboli with diameters of 1.0 mm (edge length of cubic emboli) and 1.2 mm... (Page 8)

Comment 14:

On pages 9 and 12, where descriptions for Fig. 4 and Fig. 5 are located, reorganize the paragraphs to improve flow and readability.

Reply and corresponding changes:

Thank you very much for your reminder on this issue! We have revised the manuscript to reorganize the figure sequence and corresponding text for improved logical flow and clarity.

Other corrections

To demonstrate the expandability of this study, the proposed synthesis strategy was further extended to the preparation of other metal (Hf/Ta/Yb)-doped BioGlue

(Fig. S3). Dynamic oscillation time sweep measurements showed that exhibit similar G'/G'' ratios, indicating that doping with metal elements does not affect the rheological properties of the adhesives obviously (Fig. S4). (Page 6)

Synthesis of Hf-, Ta-, Yb-doped BioGlue. To synthesize Hf-, Ta-, and Yb-doped BioGlue, HfCl_4 , TaCl_5 , and YbCl_3 were individually mixed and stirred with BSA solution to obtain metal probe/BSA mixtures containing 150 mM of the metal element and 450 mg/mL of BSA, respectively. Glutaraldehyde was then added at a volume ratio of 4:1 to form the corresponding Hf-, Ta-, and Yb-doped BioGlue. (Page 14)

Revised figures

Fig. S3. Photographs and corresponding CT images of conventional BioGlue, Hf-BioGlue, Ta-BioGlue and Yb-BioGlue with the Hf, Ta, and Yb concentration of 105 mM.

Fig. S4. Dynamic oscillatory time sweep measurements of BioGlue, Hf-BioGlue, Ta-BioGlue and Yb-BioGlue.

Response to Reviewer's Comments

Response to Reviewer 2

Comments:

The authors provided additional experimental results to address part of the comments provided by the reviewers. However, the additional data did not cover all issues raised and a number of comments (for all three reviewers) were not or not accurately addressed. Unfortunately, previous questions raised on the novelty of the approach and the degree of translational advance have not been resolved.

The authors are also encouraged to reply to the provided comments only and in a short and concise manner, and exclude lengthy text sections that are not incorporated in the manuscript.

Reply and corresponding changes:

Thank you very much for your careful evaluation and thoughtful insights. To address the reviewers' concerns regarding innovation and clinical translational potential, we provide the following clarification and explanation. Regarding innovation, the innovation of this work lies in providing an early and accurate diagnostic solution for the severe complication of detached emboli caused by the clinically widely used adhesive BioGlue. Through systematic and comprehensive optimization, we identified amorphous BiOCl as a CT labeling probe that can stably label BioGlue without affecting its performance. Systematic *in vivo* results fully demonstrate that this method can achieve accurate detection of single millimeter detached emboli. Even under conditions that mimic the presence of pulmonary calcified nodules, advanced spectral CT imaging still enables accurate and interference free detection of detached emboli. This study provides an important solution for the safe use of BioGlue and is also the first report to achieve accurate detection of single millimeter scale detached emboli.

Regarding clinical translational potential, modifying and improving biomaterials or drugs that have already been used clinically is a common strategy for achieving rapid clinical translation. This study introduces a BiOCl labeling probe with good biosafety on the basis of BioGlue, therefore demonstrating good clinical translational potential. In the future, we will further conduct in depth studies of the long-term biosafety of Bi BioGlue in large animals, explore its actual clinical translational potential and promote its clinical translation.

Response to Reviewer 3

Comments:

The authors have made substantial efforts to improve this manuscript, and I believe the revised version is now suitable for publication in its current form.

Reply and corresponding changes:

Thank you very much for your positive comments.

Response to Reviewer 5

Comment 1:

In Fig. 2f, the FTIR spectrum of BioGlue needs to be supplemented to further illustrate the impact of the introduction of BiClO on BioGlue.

Reply and corresponding changes:

Thank you for your valuable comment. We measured the FTIR spectrum of BioGlue, which is included in revised Fig. 2f.

Revised figure in manuscript

Comment 2:

The color distinction in the wells of Fig. 5a should be clearer and correspond to the colors in Fig. 5d.

Reply and corresponding changes:

Thank you for your helpful comment. We have enhanced the color contrast in Fig. 5a so that the different groups are clearly distinguishable. The updated panel is now consistent with the color scheme used in Fig. 5d.

Revised figure in manuscript

Comment 3:

The authors should summarize the limitations of the study and the future directions.

Reply and corresponding changes:

Thank you for your valuable comment. A concise summary of the limitations and future prospects has been added to the last paragraph of Results and Discussion section. Although this study provides a highly sensitive strategy for detecting BioGlue emboli, the current work remains limited to a rat model. In addition, the emboli used in this study were pre-fabricated rather than naturally formed during clinical BioGlue procedures. The biosafety evaluation of Bi BioGlue is also still relatively basic. In future work, we will perform BioGlue adhesion procedures in large animals such as pigs to simulate the natural formation of emboli and to assess the detection capability of this approach under clinically relevant conditions. We will also conduct more comprehensive long-term evaluations of the biosafety of Bi BioGlue to advance its potential for clinical translation.

Revised text in manuscript:

...Although this study provides a highly sensitive strategy for detecting BioGlue emboli, the current work remains limited to a rat model. In addition, the emboli used in this study were pre-fabricated rather than naturally formed during clinical BioGlue procedures. The biosafety evaluation of Bi BioGlue is also still relatively basic. In future work, we will perform BioGlue adhesion procedures in large animals such as pigs to simulate the natural formation of emboli and to assess the detection capability of this approach under clinically relevant conditions. We will also conduct more comprehensive long-term evaluations of the biosafety of Bi BioGlue to advance its potential for clinical translation.

Comment 4:

There are inconsistencies in formatting and grammatical errors, such as the use of both “Bioglue” and “BioGlue”.

Reply and corresponding changes:

We have carefully proofread the manuscript and corrected all terminology and spelling inconsistencies to ensure full uniformity throughout the text and figures.

Response to Reviewer 6

Comments:

The authors incorporated Bi element in commercial surgical adhesives (BioGlue) for noninvasive detection of adhesive-induced emboli using spectral CT imaging. The results clearly demonstrated that the detecting time is over than 40 days, and the size could limit to 1.2 mm. The authors have carefully revised all the comments. So, in my opinion, the revision should be published as it is now.

Reply and corresponding changes:

Thank for your such positive comments.